# FullStack-Agent: Enhancing Agentic Full-Stack Web Coding via Development-Oriented Testing and Repository Back-Translation

**Zimu Lu** [* 1]  **Houxing Ren** [* 1]  **Yunqiao Yang** [1]  **Ke Wang** [1]  **Zhuofan Zong** [1]  **Mingjie Zhan** [† 1 2]  **Hongsheng Li** [† 1 3 4]

## Abstract

Assisting non-expert users to develop complex interactive websites has become a popular task for LLM-powered code agents. However, existing code agents tend to only generate frontend web pages, masking the lack of real full-stack data processing and storage with fancy visual effects. Notably, constructing production-level full-stack web applications is far more challenging than only generating frontend web pages, demanding careful control of data flow, comprehensive understanding of constantly updating packages and dependencies, and accurate localization of obscure bugs in the codebase. To address these difficulties, we introduce *FullStack-Agent*, a unified agent system for full-stack agentic coding that consists of three parts: (1) *FullStack-Dev*, a multi-agent framework with strong planning, code editing, codebase navigation, and bug localization abilities. (2) *FullStack-Learn*, an innovative data-scaling and self-improving method that back-translates crawled and synthesized website repositories to improve the backbone LLM of FullStack-Dev. (3) *FullStack-Bench*, a comprehensive benchmark that systematically tests the frontend, backend and database functionalities of the generated website. Our FullStack-Dev outperforms the previous state-of-the-art method by 8.7%, 38.2%, and 15.9% on the frontend, backend, and database test cases respectively. Additionally, FullStack-Learn raises the performance of a 30B model by 9.7%, 9.5%, and 2.8% on the three sets of test cases through self-improvement, demonstrating the effectiveness of our approach.

[1]Multimedia Laboratory (MMLab), The Chinese University of Hong Kong [2]SenseTime Research [3]Shenzhen Loop Area Institute [4]Ace Robotics. Correspondence to: Zimu Lu <luzimu@link.cuhk.edu.hk>, Mingjie Zhan <zhanmingjie@sensetime.com>, Hongsheng Li <hsli@ee.cuhk.edu.hk>.

*Proceedings of the $43^{rd}$ International Conference on Machine Learning*, Seoul, South Korea. PMLR 306, 2026. Copyright 2026 by the author(s).

## 1. Introduction

Assisting non-expert users to develop complicated web applications based on natural language instructions has become a popular task for Large Language Model (LLM)-powered (Qwen, 2025b; Wang et al., 2025). Various commercial products[1] and research studies (Lu et al., 2025a; Wan et al., 2025) have provided solutions to this task through agentic coding, code execution, and GUI-agent–based testing. However, these systems tend to generate frontend-only websites even when a backend and data storage are needed to fully support the functionality required by user instructions. They often mask the lack of real data flow with fancy visual effects to create an appearance of interactivity. For example, in one generated website, while a form can be submitted and a success notice appears, no data would actually be processed or stored due to the lack of a backend and database implementation. Additionally, many of these methods generate only an HTML file (Xiao et al., 2025a; Si et al., 2025) or a very simple codebase (Lu et al., 2025a), lacking scalability for production environments.

However, full-stack websites are rather complicated, making it hard to design effective generation methods. There are at least three challenges in building code agents capable of generating production-grade full-stack websites: (1) Real-world web development frameworks, such as Next.js and NestJS, involve large, complex codebases, requiring efficient code navigation and accurate localization and correction of obscure errors. (2) The complicated workflow of full-stack coding demands long-term reasoning, skillful tool invocation, and expert mastery of web packages–areas in which current backbone LLMs still have considerable room for improvement. (3) Evaluating full-stack website generation remains challenging, as existing GUI-agent-based benchmarks such as WebGen-Bench (Lu et al., 2025b) primarily judge UI-level interactions and fail to detect missing or incorrect backend implementations.

To address these challenges, we introduce FullStack-Agent, a unified system for full-stack website generation that aims to close the gap between real-world web development and current agentic approaches. It jointly advances the effective-

---
[1]https://bolt.new, https://lovable.dev

ness of full-stack development workflows, the agentic coding ability of backbone LLMs, and the comprehensiveness of website evaluation by proposing *three* tightly coupled components that work together to enable scalable and verifiable full-stack website construction. The three components, covering the agent framework, backbone LLM training, and full-stack evaluation, are detailed below:

**FullStack-Dev** To effectively coordinate the complicated development workflow of full-stack website generation, we propose *FullStack-Dev*, a multi-agent system that takes inspiration from real-world development processes. A planning agent, serving as the lead architect, designs the structure of the full-stack website, and delegates frontend and backend plans to the corresponding coding agents. The two coding agents serve as frontend and backend engineers, and are equipped with efficient code editing, shell command execution, and website debugging tools, allowing them to dynamically control the coding process. In particular, the specially designed frontend and backend debugging tools can efficiently locate and correct subtle errors, greatly enhancing the coding agents' development abilities.

**FullStack-Learn** Even with a powerful agent framework, the agentic coding skills and expert knowledge possessed by the backbone LLM are still crucial to the overall performance of the system. Therefore, we introduce *FullStack-Learn*, a data-scaling and model self-improvement method that generates high-quality agent trajectories through augmentation and back-translation of website repositories collected from GitHub. Our artful design combines global planning and information gathering with local code implementation, effectively solving the non-trivial problem of converting a complicated repository into agent trajectories that implement it from scratch. These agent trajectories, generated from real-world codebases and used for supervised fine-tuning, enable the LLMs to learn valuable agentic coding abilities and expert understanding of website development frameworks.

**FullStack-Bench** Existing website evaluation benchmarks such as WebGen-Bench (Lu et al., 2025b) focus on frontend reactions observed by a GUI-agent judge, failing to detect false positive cases that show correct frontend effects yet lack a real backend implementation. To solve this problem, we introduce *FullStack-Bench*, a full-stack evaluation benchmark that tests frontend, backend, and database functionalities by leveraging agent judges to run multiple carefully constructed test cases on each website. During frontend and backend tests, database logs are also gathered and evaluated to ensure that the actions are accompanied by adequate interactions with the database.

Together, the three components of the FullStack-Agent sys-

tem comprehensively address the challenges of production-level full-stack development and significantly improve performance on full-stack code generation. Extensive experiments demonstrate the effectiveness of our approach. Testing FullStack-Dev with Qwen3-Coder-480B-A35B-Instruct as the backbone LLM on FullStack-Bench results in accuracies of 64.7%, 77.8%, and 77.9% in frontend, backend, and database test cases respectively, outperforming the previous state-of-the-art method by 8.7%, 38.2%, and 15.9%. Additionally, training Qwen3-Coder-30B-Instruct with FullStack-Learn improves its accuracy by 9.7%, 9.5%, and 2.8% in the three sets of test cases respectively, demonstrating the effectiveness of our training method.

In summary, our contributions are as follows:

- We introduce *FullStack-Dev*, a multi-agent full-stack development framework with highly effective coding tools, significantly outperforming the previous state-of-the-art method.

- We propose *FullStack-Learn*, an iterative self-improvement method that substantially improves the full-stack development ability of the backbone LLM through repository augmentation and back-translation.

- We construct *FullStack-Bench*, a novel benchmark that comprehensively evaluates the functionalities of the generated full-stack websites.

## 2. FullStack-Agent

In this section, we introduce *FullStack-Agent*, a unified system for agentic full-stack development that consists of a multi-agent framework for full-stack generation, an iterative self-improvement pipeline for backbone LLM training, and a comprehensive benchmark for full-stack evaluation.

### 2.1. FullStack-Dev

To address the challenges of full-stack development, such as constructing efficient data flows, managing complex file structures, and debugging obscure bugs, we propose *FullStack-Dev*, a multi-agent framework with powerful tools for dynamic coding and debugging. As shown in the left part of Fig. 1, it starts with the *Planning Agent*, which generates high-level frontend and backend development plans, much like a lead full-stack architect. The plans are sent to the coding agents, which serve as frontend and backend engineers. Then the *Backend Coding Agent* starts developing the backend, and provides a summary of the APIs it constructed. Finally, the *Frontend Coding Agent* creates the frontend based on the backend APIs. The agents work in sandbox environments to ensure isolation, safety, and execution stability.

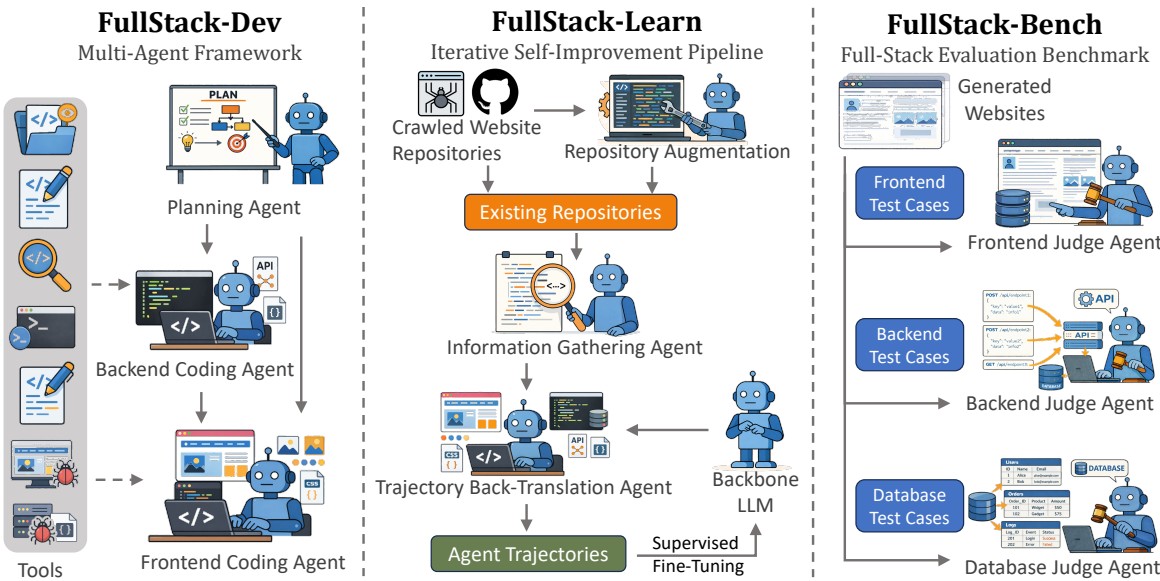

**FullStack-Dev**
Multi-Agent Framework

Planning Agent

Backend Coding Agent

Tools

Frontend Coding Agent

**FullStack-Learn**
Iterative Self-Improvement Pipeline

Crawled Website
Repositories

Repository Augmentation

Existing Repositories

Information Gathering Agent

Trajectory Back-Translation Agent

Backbone
LLM

Agent Trajectories

Supervised
Fine-Tuning

**FullStack-Bench**
Full-Stack Evaluation Benchmark

Generated
Websites

Frontend
Test Cases

Frontend Judge Agent

Backend
Test Cases

Backend Judge Agent

Database
Test Cases

Database Judge Agent

*Figure 1.* The FullStack-Agent system. It combines a multi-agent development framework equipped with efficient coding and debugging tools (FullStack-Dev), an iterative self-improvement method that enhances LLMs through repository augmentation and back-translation (FullStack-Learn), and a comprehensive benchmark evaluating frontend, backend, and database functionalities (FullStack-Bench).

**Planning Agent.** The Planning Agent takes the user instruction and creates a high-level frontend and backend development plan that focuses on describing the page layouts, components, and data flow of the frontend as well as the entities and API endpoints used by the backend. It outputs the frontend and backend designs in JSON format, which is both easy to parse and straightforward for the downstream coding agents to understand. All data structures in the plans are defined down to the most granular types (e.g., integer), ensuring the smooth data flow among the frontend, the backend, and the database.

**Coding Agents.** The Backend Coding Agent and the Frontend Coding Agent run sequentially to implement the plans generated by the Planning Agent. Both agents are equipped with efficient tools, including general coding tools such as code reading, file writing, string replacement, searching, and shell command execution, as well as two specialized debugging tools for the frontend and backend. These tools cover all the resources needed by a website engineer, enabling the agents to dynamically control the full-stack development workflow. In particular, we describe the two specially designed debugging tools as follows:

- **Frontend Debugging Tool.** The Frontend Debugging Tool takes a test instruction, automatically starts the website service, and runs a GUI-agent debugging process based on the given instruction. As the GUI agent interacts with the website, the debugging tool closely monitors the terminal and browser console outputs. When an error is detected, it sends a message to the GUI agent, asking the GUI agent to identify which action led to the error

and to return an error analysis along with the corresponding error messages to the Coding Agent. This debugging mechanism monitors error messages as well as website reactions, allowing for more efficient localization of the problem; in contrast, previous GUI-agent-based debugging methods (Lu et al., 2025a; Wan et al., 2025) rely on blind interaction with the website and only produce coarse-grained observations. Also, unlike previous methods, which repeatedly run test cases pre-defined at the beginning of the coding process, our tool dynamically generates test cases based on the current state of development, which provides better control of the workflow.

- **Backend Debugging Tool.** The Backend Debugging Tool takes the URL, the request method (e.g., POST), and the request data, and automatically starts the service, makes the request, and returns the response message as well as the outputs from the backend console, similar to how a backend developer works with API debugging tools such as Postman (Postman, 2026). This process is much more efficient than using shell commands to test APIs, a process that would require multiple steps to achieve the same effect and is therefore prone to mistakes. Removing the Backend Debugging Tool results in an increase in the Backend Coding Agent's average number of iterations from 74.9 to 115.5, which demonstrates the crucial role the tool plays in improving development efficiency.

In summary, FullStack-Dev leverages a multi-agent mechanism to facilitate the coordination of planning, frontend coding, and backend coding. The design of efficient tools, especially the two debugging tools, enables the Coding

Agents to dynamically control the development workflow, making the framework highly effective. The agent prompts are provided in Appendix H, while the configuration of the tools is described in Appendix A.

## 2.2. FullStack-Learn

Even with a powerful framework, the agentic coding ability and expert knowledge of the backbone LLM remain essential to the overall performance of the system. Therefore, we propose *FullStack-Learn*, a robust iterative self-improvement pipeline that enhances the capabilities of the backbone LLM through repository augmentation and back-translation.

**Repository Back-Translation.** To effectively utilize high-quality real-world website repositories crawled from the internet, we introduce Repository Back-Translation, a data generation scheme that transforms existing repositories into agentic trajectories. As illustrated in the middle part of Fig. 1, first, the *Information Gathering Agent* reads through important files in the repository to produce a general summary. Then, the summary is provided to the *Trajectory Back-Translation Agent* to guide it through the process of transcribing the existing repository into an empty template. The trajectory produced by the Trajectory Back-Translation Agent is then transformed by a rule-based program that purges all traces of the original repository from it, resulting in a cleaned agentic trajectory. The transformation process is detailed in Algorithm 2 of Appendix C.

- **Information Gathering Agent.** The Information Gathering Agent leverages its directory navigation, file reading, and code searching tools to understand the structure and functionality of the repository. It then outputs a summary consisting of a description of the repository, a quality score, the frontend and backend plans corresponding to the repository, as well as a user instruction that would plausibly result in such a repository. The plans and the user instruction are later used to guide the construction of the agent trajectories, while the quality score is used to filter out low-quality repositories.

- **Trajectory Back-Translation Agent.** The Trajectory Back-Translation Agent is provided with the user instruction and a set of high-level plans, as well as a workspace containing the original repository and an empty template of the same web framework (e.g., Next.js) as the original repository. It is tasked with reproducing the contents of the original repository in the template. As the transcription is guided by the high-level plans, the order of actions is very close to that of direct development, which usually includes first reading the relevant files and then implementing the functionalities. This means that the resulting transcription trajectories can be easily transformed

into development trajectories with a rule-based program without compromising their logical coherence.

The coordinated operation of the Information Gathering Agent and the Trajectory Back-Translation Agent effectively produces large-scale, high-quality agent trajectories for SFT training, enabling models to learn from trajectories derived from existing website repositories. Detailed prompts are presented in Fig. 14 and Fig. 15 of Appendix I.

**Repository Augmentation.** To further scale the number of high-quality agent trajectories, we propose Repository Augmentation, which produces a large number of synthesized repositories by implementing numerous augmentations on existing repositories. Augmenting an existing repository is simpler than generating a new codebase from scratch, as it requires less coding and can draw on the existing code as reference. We perform Repository Augmentation by first generating several possible augmentation plans with the *Augmentation Planning Agent*, and then implementing the plans separately using the *Augmentation Implementing Agent*. The Repository Augmentation process creates five times the number of original repositories, greatly scaling the number of generated trajectories. The prompts for the two agents are presented in Fig. 16 of Appendix I.

- **Augmentation Planning Agent.** The Augmentation Planning Agent first goes through the original repository to understand its structure and functionality, then creates five suggested augmentation plans: one simplification plan, one extension plan, and three application transition plans. The application transition plans propose alternative applications whose codebases have a structure similar to that of the original repository.

- **Augmentation Implementing Agent.** The Augmentation Implementing Agent takes a plan created by the Augmentation Planning Agent and implements it in the original repository. It also runs the debugging tools on the augmented repository to fix any breaking issues and ensure the quality of the synthetic repository. At the end of the trajectory, the agent is asked to verify whether all the changes required by the augmentation plan have been fully implemented based on the past agent messages and the state of the codebase, and only the samples that pass the verification are retained. The verification prompt is presented in Fig. 17 of Appendix I.

**Iterative Self-Improvement.** We introduce an iterative self-improvement training process to effectively leverage the data generated with Repository Back-Translation and Repository Augmentation. As shown in Algorithm 1, first, we generate an initial dataset, $D_0$, with the original LLM, $M_0$ based on repositories crawled from GitHub using Repository Back-Translation. $D_0$ is relatively small due to the limited full-stack coding ability of $M_0$. We proceed to train

**Algorithm 1** Iterative Self-Improvement

**Input:** initial backbone model $M_0$; real-world repositories $\mathcal{R}_{\text{real}}$; back-translation operator $\mathcal{B}(\cdot)$; augmentation operator $\mathcal{A}(\cdot)$; model-updating procedure $\mathcal{U}_{\text{SFT}}(\cdot, \cdot)$
**Output:** improved backbone model $M_{\text{final}}$
**Round 1: Initial trajectory generation**
$D_0 \leftarrow \mathcal{B}(\mathcal{R}_{\text{real}}, M_0)$
$M_1 \leftarrow \mathcal{U}_{\text{SFT}}(M_0, D_0)$
**Round 2: Data scaling via augmentation and back-translation**
$\mathcal{R}_{\text{aug}} \leftarrow \mathcal{A}(\mathcal{R}_{\text{real}}, M_0)$
$D_{\text{aug}} \leftarrow \mathcal{B}(\mathcal{R}_{\text{aug}}, M_1)$
$D_1 \leftarrow D_0 \cup D_{\text{aug}}$
**Step 3: Final training**
$M_{\text{final}} \leftarrow \mathcal{U}_{\text{SFT}}(M_0, D_1)$

*Table 1.* Counts of frontend, backend, and database test cases as well as user instructions in FullStack-Bench. Each user instruction corresponds to multiple test cases in the three aspects.

| Name | Frontend | Backend | Database | User Instructions |
|------|----------|---------|----------|-------------------|
| Value | 647 | 604 | 389 | 101 |

$M_0$ with $D_0$, resulting in an intermediate model $M_1$. Then, using $M_1$, we scale the training data by back-translating synthetic repositories created by $M_0$ using Repository Augmentation. The newly generated data, combined with $D_0$, results in $D_1$. Training $M_0$ on $D_1$ results in the final model $M_{\text{final}}$, which possesses much stronger full-stack development abilities. The iteration process is also shown in the lower half of the FullStack-Learn part of Fig. 1.

### 2.3. FullStack-Bench

Current website development benchmarks, such as WebGen-Bench (Lu et al., 2025b), focus mainly on frontend interactions, failing to effectively verify the correctness of backend and database implementations. To solve this problem, we construct *FullStack-Bench*, a novel benchmark that leverages comprehensive test cases for the frontend, backend, and database to evaluate the websites generated based on natural-language user instructions. The statistics of the test cases are shown in Tab. 1. The 101 website generation user instructions used in FullStack-Bench are taken from WebGen-Bench (Lu et al., 2025b), along with 647 frontend test cases, while the 604 backend test cases and 389 database test cases are newly constructed through a combination of LLM annotation and human refinement. Prompts for the frontend, backend, and database testing are presented in Appendix J. The three levels of testing are described below:

**Frontend Test.** The frontend test cases are carried out by a GUI-agent judge, similar to WebGen-Bench (Lu et al.,

2025b). Unlike Lu et al. (2025b), which uses Qwen2.5-VL-32B-Instruct (Bai et al., 2025b) to power the GUI agent, we use the new Qwen3-VL-235B-A22B-Instruct (Bai et al., 2025a) for enhanced accuracy when facing more complicated websites. To validate the data flow accompanying each frontend test case, we extract database log entries that are written during the frontend interaction, and append a message at the end of the GUI-agent testing session to verify whether the database log entries correctly reflect the database interactions needed by the frontend actions. As defined in Lu et al. (2025b), frontend results can be YES, PARTIAL, or NO, and the accuracy is computed as Accuracy $= \frac{N_{\text{Yes}} + 0.5 \times N_{\text{Partial}}}{N_{\text{Total}}} \times 100\%$, with $N_{\text{Yes}}$, $N_{\text{Partial}}$ and $N_{\text{Total}}$ denoting the number of YES, PARTIAL, and total test cases, respectively. The difference is that in FullStack-Bench, the YES and PARTIAL outputs from the GUI-agent judge are counted only if the database interaction check passes. Following Lu et al. (2025b), we also compute the Appearance Score to evaluate the website appearance.

**Backend Test.** In backend testing, the judge agent is first instructed to gather information about all the backend API endpoints. Then, a new message is appended after the information-gathering steps, instructing the judge agent to make requests to validate the functionality of the current test case. The judge agent sends requests to related APIs, decides whether the functionality is fulfilled based on the responses, and outputs YES or NO. Qwen3-Coder-480B-A35B-Instruct (Qwen, 2025a) is used as the agent backbone. As each website corresponds to multiple backend test cases, the API information gathering steps are reused by all the backend test cases to avoid a waste of computation. The accuracy is computed as the number of YES cases divided by the total number of test cases.

**Database Test.** To test the database structure, we first extract all the column names and the first five rows from each table in the database to create a snapshot. Placing it in JSON format, we ask the judge agent to decide whether the data requirement in the test case has been fulfilled based on the database snapshot. The judge agent outputs YES or NO, and the accuracy is computed as the number of YES cases divided by the total number of test cases.

## 3. Experiments

In this section, we present experimental results for both FullStack-Dev and FullStack-Learn, along with ablation studies to demonstrate the effectiveness of our approach.

### 3.1. FullStack-Dev Results

**Test Settings.** We test FullStack-Dev with Qwen3-Coder-30B-A3B-Instruct and Qwen3-Coder-480B-A35B-

*Table 2.* Evaluation results of FullStack-Dev on FullStack-Bench compared to other popular agentic coding frameworks. FE: Frontend. BE: Backend. DB: Database. "w/ Valid DB": with valid database interaction logs. All values are percentages except the Appearance Score, which scales from 1-5. The highest values are in **bold**, while the second highest values are underlined.

| Framework | Model | FE Acc. | FE Acc. w/ Valid DB | BE Acc. | BE Acc. w/ Valid DB | DB Acc. | Appearance Score |
|---|---|---|---|---|---|---|---|
| WebGen-Agent | Qwen3-Coder-480B-A35B-Inst. | 61.4 | 56.0 | 67.2 | 39.6 | 62.0 | 3.63 |
| TDDev | Qwen3-Coder-480B-A35B-Inst. | 23.3 | 21.8 | 41.2 | 10.8 | 28.5 | 1.96 |
| OpenHands | Qwen3-Coder-480B-A35B-Inst. | 45.1 | 40.4 | 64.9 | 35.8 | 72.5 | 2.66 |
| Bolt.diy | Qwen3-Coder-480B-A35B-Inst. | 22.8 | 20.3 | 35.9 | 12.7 | 13.6 | 1.96 |
| Qwen-Code | Qwen3-Coder-480B-A35B-Inst. | 39.9 | 36.2 | 59.3 | 25.8 | 68.4 | 2.40 |
| **FullStack-Dev** | Qwen3-Coder-480B-A35B-Inst. | **67.8** | **64.7** | **83.4** | **77.8** | **77.9** | **3.72** |
| WebGen-Agent | Qwen3-Coder-30B-A3B-Inst. | 39.5 | 35.8 | 25.0 | 16.6 | 14.4 | 2.49 |
| TDDev | Qwen3-Coder-30B-A3B-Inst. | 13.7 | 12.5 | 13.4 | 8.8 | 6.9 | 1.21 |
| OpenHands | Qwen3-Coder-30B-A3B-Inst. | 21.5 | 19.4 | 26.7 | 16.7 | 22.6 | 1.62 |
| Bolt.diy | Qwen3-Coder-30B-A3B-Inst. | 5.9 | 5.0 | 5.6 | 3.6 | 13.1 | 0.46 |
| Qwen-Code | Qwen3-Coder-30B-A3B-Inst. | 7.0 | 6.0 | 20.9 | 9.9 | 18.3 | 0.75 |
| **FullStack-Dev** | Qwen3-Coder-30B-A3B-Inst. | **41.4** | **37.2** | **45.5** | **38.7** | **50.9** | **2.97** |
| WebGen-Agent | DeepSeek-V3 | 48.8 | 44.4 | 64.2 | 59.6 | 71.7 | 3.33 |
| TDDev | DeepSeek-V3 | 41.3 | 39.3 | 61.3 | 57.0 | 72.5 | 2.28 |
| OpenHands | DeepSeek-V3 | 21.8 | 19.3 | 76.5 | 74.0 | 74.6 | 1.69 |
| Bolt.diy | DeepSeek-V3 | 25.4 | 23.0 | 62.6 | 32.1 | 65.3 | 2.11 |
| Qwen-Code | DeepSeek-V3 | 22.1 | 20.1 | 65.9 | 64.6 | 70.7 | 1.58 |
| **FullStack-Dev** | DeepSeek-V3 | **73.6** | **69.6** | **91.1** | **89.7** | **79.2** | **4.02** |

Instruct (Qwen, 2025a) as the backbone LLMs. During inference, greedy decoding is used, with a context length of 131,072. The maximum number of tool calls is 400. In frontend and backend evaluations, the accuracies that require correct database interactions (FE Acc. w/ Valid DB and BE Acc. w/ Valid DB in Tab. 2) are the main metrics, though we also report accuracies that ignore database interactions, shown in gray text. When mentioning frontend and backend accuracy, we are referring to the accuracy that considers database interactions unless otherwise specified.

**Baselines.** We choose website development agents including WebGen-Agent (Lu et al., 2025a), TDDev (Wan et al., 2025), and Bolt.diy (stackblitz labs, 2024), as well as general coding agents including OpenHands (Wang et al., 2025) and Qwen-Code (Qwen, 2025b), as baselines. These agents tend to only generate the frontend web pages when provided with only user instructions, so we also explicitly prompt them to generate backend components as well. Details of the baseline implementations are described in Appendix E.

**Results.** As shown in Tab. 2, FullStack-Dev with Qwen3-Coder-480B-A35B-Instruct results in the highest accuracies of 64.7%, 77.8% and 77.9% in frontend, backend, and database tests, respectively, outperforming the previous state-of-the-art method, WebGen-Agent (with accuracies of 56.0%, 39.6%, and 62.0%) by 8.7%, 38.2%, and 15.9%, respectively. With Qwen3-Coder-30B-A3B-Instruct and DeepSeek-V3, FullStack-Dev also results in the highest accuracies in the frontend, backend, and database tests, respectively, demonstrating the effectiveness of our approach.

In particular, DeepSeek-V3 achieves the best performance overall, showing the generalizability of our pipeline beyond the Qwen family. Our method also achieves the highest appearance score, which could be attributed to the frontend debugging tool's ability to adjust rendering issues. Notably, for most of the baseline methods, the backend accuracies are much lower than the frontend accuracies, showing that even with explicit requests to generate the backend, these methods still tend to focus on the frontend, often using mock data to serve as the backend. In contrast, our FullStack-Dev method has backend accuracies higher than the frontend accuracies, showing that our full-stack websites mostly possess functional backends. Error analysis of the generated websites is presented in Appendix F.

### 3.2. FullStack-Learn Results

**Training and Inference Settings.** During Repository Back-Translation, the backbone LLMs of both the Information Gathering Agent and the Trajectory Back-Translation Agent have a temperature of 0.5 and a context length of 131,072. The generated trajectories are filtered based on outputs of the debugging tools, as detailed in Appendix D. We conduct the iterative self-improvement process with Qwen3-Coder-30B-A3B-Instruct. In the first round, we use Qwen3-Coder-30B-A3B-Instruct as the backbone LLM to generate a dataset of 2K trajectories based on repositories crawled from GitHub, and train the LLM on this dataset, resulting in a model denoted as FullStack-Learn-LM-round1. In the second round, we use FullStack-Learn-LM-round1 as the backbone LLM to generate another 8K trajectories based on

*Table 3.* FullStack-Bench results of two rounds of training on Qwen3-Coder-30B-A3B-Instruct using FullStack-Learn. FE: Frontend. BE: Backend. DB: Database."w/ Valid DB": with valid database interaction logs. All values are percentages except the Appearance Score.

| Test Name | Data | FE Acc. | FE Acc. w/ Valid DB | BE Acc. | BE Acc. w/ Valid DB | DB Acc. | Appearance Score |
|---|---|---|---|---|---|---|---|
| Qwen3-Coder-30B-A3B-Inst. | – | 41.4 | 37.2 | 45.5 | 38.7 | 50.9 | 2.97 |
| FullStack-Learn-LM-round1 | Crawled 2K | 49.3 | 42.3+5.1 | 55.6 | 45.4+6.7 | 51.2+0.3 | 3.32+0.35 |
| FullStack-Learn-LM-round2 | Crawled 2K + Agemented 8K | **52.5** | **46.9**+9.7 | **57.6** | **48.2**+9.5 | **53.7**+2.8 | **3.40**+0.43 |

augmented repositories, and train Qwen3-Coder-30B-A3B-Instruct on the combined 10K trajectories from both rounds, resulting in FullStack-Learn-LM-round2. The generated trajectories are decontaminated against FullStack-Bench by comparing their user instructions and filtering out those with 5-gram Jaccard similarity scores larger than 0.6 and cosine similarities between sentence embeddings (Reimers & Gurevych, 2019) larger than 0.7. In both rounds, the models are trained for 2 epochs, with a learning rate of 2e-5 and a batch size of 32 on 32 H800 GPUs. We test the FullStack-Learn-LM models on FullStack-Dev, with a temperature of 0 and a context length of 131,072.

**Results.** As shown in Tab. 3, the two rounds of training consistently improve the accuracy across the frontend, backend, database, and appearance tests. After the two rounds of training, the FullStack-Learn-LM achieves an accuracy of 46.9%, 48.2%, and 53.7% in frontend, backend, and database tests, respectively, outperforming the original Qwen3-Coder-30B-A3B-Instruct by 9.7%, 9.5%, and 2.8%, which demonstrates the effectiveness of our FullStack-Learn method. Notably, the whole process is entirely self-improving, without relying on any stronger model, which suggests that our method can potentially generalize to larger and stronger models.

### 3.3. Ablation Studies

In this section, we present ablation studies on the design choices for FullStack-Dev and the data generation methods for FullStack-Learn, and we analyze the reliability of FullStack-Bench 's test results.

**Analysis of FullStack-Dev Design.** We analyze the contribution of the multi-agent mechanism, the Frontend Debugging Tool, and the Backend Debugging Tool to the performance of FullStack-Dev by removing them one by one and testing the results. As shown in Tab. 4, first, removing the multi-agent mechanism reduces the accuracy across all metrics, as a single agent fails to effectively coordinate the full-stack development task. Second, removing the Backend Debugging Tool has a larger effect on the backend accuracy, while removing the Frontend Debugging Tool has a larger effect on the frontend accuracy, which is consistent with their respective roles. Removing both debugging tools results in

considerable degradation in both frontend and backend.

**Analysis of FullStack-Learn Data Generation Method.** To analyze the effect of Repository Back-Translation, we generate 2K trajectories directly from user instructions randomly sampled from WebGen-Instruct (Lu et al., 2025b), and compare the performance of the resulting model to that of FullStack-Learn-LM-round1, which is trained with 2K trajectories generated with Repository Back-Translation. As shown in Tab. 5, training on trajectories generated with Repository Back-Translation significantly increases the accuracies and the appearance score, while training on the directly-generated trajectories fails to notably improve the overall performance. This significant performance gap is likely due to the fact that our approach enables the LLM to learn from high-quality, real-world repositories.

**Analysis of Evaluation Reliability.** To analyze the reliability of the testing pipeline of FullStack-Bench, we randomly sample 200 instances from the frontend, backend, and database samples each. We then ask four student volunteers with computer-science-related bachelor's degrees to manually check their correctness, annotating a sample as correct only when the evaluation trajectory and database interaction logs fully support the final result. The human alignment accuracy is computed as the percentage of samples judged as correct by human annotators. As shown in Tab. 6, the accuracies of the frontend, backend, and database are all above 90%, demonstrating the reliability of the testing pipeline. Details of the human annotation, including the annotation interface and guidelines are presented in Appendix G.

**Analysis of Sensitivity to Judge Models.** To evaluate the sensitivity of FullStack-Bench results to the choice of judge model, we replaced Qwen3-Coder-480B with DeepSeek-V3. As shown in the Tab. 7, replacing the judge model does not significantly affect the scores. DeepSeek-V3's results are consistently lower by about 1-2 points, and the rankings of the tested agents remain unchanged.

**Computational Costs.** The computational costs of FullStack-Dev with Qwen3-Coder-480B are presented in the Tab. 8. While the token count and runtime are slightly higher than those of baseline methods, they are still within an acceptable range. This is a necessary trade-off consid-

*Table 4.* Evaluation of different ablation settings of FullStack-Dev using Qwen3-Coder-480B-A35B-Instruct. FE: Frontend. BE: Backend. DB: Database. "w/ Valid DB": with valid database interaction logs. All values are percentages except the Appearance Score.

| Test Name | FE Acc. | FE Acc. w/ Valid DB | BE Acc. | BE Acc. w/ Valid DB | DB Acc. | Appearance Score |
|---|---|---|---|---|---|---|
| FullStack-Dev w/o Multi-Agent Mechanism | 66.5 | 62.1 | 78.0 | 61.4 | 63.5 | 3.43 |
| FullStack-Dev w/o Backend Debugging Tool | 65.5 | 62.1 | 71.7 | 57.9 | 76.6 | 3.23 |
| FullStack-Dev w/o Frontend Debugging Tool | 55.1 | 51.9 | 79.0 | 76.6 | 75.6 | 3.17 |
| FullStack-Dev w/o Either Debugging Tools | 54.0 | 51.0 | 71.7 | 61.9 | 74.8 | 3.01 |
| **FullStack-Dev w/ Both Debugging Tools** | **67.8** | **64.7** | **83.4** | **77.8** | **77.9** | **3.72** |

*Table 5.* Ablation of the Repository Back-Translation method using Qwen3-Coder-30B-A3B-Instruct. FE: Frontend. BE: Backend. DB: Database. "w/ Valid DB": with valid database interaction logs. All values are percentages except the Appearance Score.

| Test Name | FE Acc. | FE Acc. w/ Valid DB | BE Acc. | BE Acc. w/ Valid DB | DB Acc. | Appearance Score |
|---|---|---|---|---|---|---|
| No Training | 41.4 | 37.2 | 45.5 | 38.7 | 50.9 | 2.97 |
| Trained on Directly-Generated 2K | 40.2 | 36.2 | 50.7 | 33.6 | 47.8 | 2.87 |
| **Trained on Back-Translated 2K** | **49.3** | **42.3** | **55.6** | **45.4** | **51.2** | **3.32** |

*Table 6.* Human alignment accuracies for frontend, backend, and database tests on 200 randomly sampled test instances.

| Test Name | Frontend | Backend | Database |
|---|---|---|---|
| Human Alignment (%) | 90.5 | 94.0 | 97.5 |

ering the complexity of the full-stack development task, and the improved performance of our method justifies the additional computational cost and runtime.

# 4. Related Work

**Website Development Agents and Pipelines.** Website development has become a popular task for code agents, and various methods have been proposed. Among them, MR-Web (Wan et al., 2024) only generates HTML and CSS files. Others, such as Bolt.diy (stackblitz labs, 2024), WebGen-Agent (Lu et al., 2025a), and TDDev (Wan et al., 2025) generate relatively simple codebases that contain little to no backend or database implementation unless specially prompted. Unlike FullStack-Dev, they lack dynamic code navigation methods, and instead cram all the code into the context window, limiting their ability to work on complicated codebases. Similar to the Frontend Debugging Tool of FullStack-Dev, WebGen-Agent and TDDev also use a GUI agent to provide feedback, though their test cases are pre-defined at the start of the generation, and the GUI agents blindly interact with the website, whereas our method supports dynamic creation of test cases and accurate localization of errors. General code agents (Wang et al., 2025; Qwen, 2025b) also tend to only generate the frontend, and without specialized feedback and system instructions, they exhibit lower performance compared to specialized agents.

**Website Development Benchmarks.** Existing website development benchmarks mostly focus on evaluating the frontend appearance and functionalities. Many of them (Si et al., 2025; Yun et al., 2024; Guo et al., 2025; Xiao et al., 2025b;a; Sun et al., 2025; Zhang et al., 2025a) only evaluate the generation of simple HTML files based on given design images, which can be achieved by the MLLMs alone without the need for agentic systems. Web-Bench (Xu et al., 2025) evaluates the website code generation ability of LLMs by running them on fixed pipelines, which fails to evaluate the ability of agentic systems. WebGen-Bench (Lu et al., 2025b) evaluates agentic coding of multi-file codebases with numerous functionality requirements, yet it judges the websites based solely on GUI-agent interaction results, failing to adequately test the backend and database implementations. In contrast, our FullStack-Bench evaluates the frontend, backend, and database implementations with comprehensive test cases and agent-based judges, effectively evaluating the full-stack websites created by agentic coding systems.

**Training Methods to Improve Software Development.** Various training methods have been proposed to improve the software development abilities of LLMs. WebCode2M (Gui et al., 2025) and WebSight (Laurençon et al., 2024) use screenshot and HTML pairs in SFT to improve MLLMs' ability to convert images to HTML code. Various reinforcement learning methods (Wei et al., 2025; Ma et al., 2025; Zhang et al., 2025b) and SFT methods (Yang et al., 2025; Pan et al., 2025) have been introduced for the task of fixing GitHub issues. However, this task focuses on generating a patch to an existing codebase, which is fundamentally different from generating full-stack websites from scratch, so these methods do not apply to our task. WebGen-Bench (Lu et al., 2025b) relies on a larger LLM to generate website development trajectories from synthetic user instructions to

*Table 7.* Evaluation of FullStack-Bench sensitivity to the choice of judge model. Replacing Qwen3-Coder-480B with DeepSeek-V3 produces consistently slightly lower scores, typically by about 1–2 points, while preserving the ranking of the tested agents.

| Agent | Judge Model | Frontend Acc. w/ Valid DB | Backend Acc. w/ Valid DB | Database Acc. |
|---|---|---|---|---|
| FullStack-Agent | Qwen3-Coder-480B | 69.6 | 89.7 | 79.2 |
| FullStack-Agent | DeepSeek-V3 | 67.5 | 87.1 | 78.4 |
| WebGen-Agent | Qwen3-Coder-480B | 44.4 | 59.6 | 71.7 |
| WebGen-Agent | DeepSeek-V3 | 42.0 | 58.8 | 71.0 |

*Table 8.* Computational costs of FullStack-Dev with Qwen3-Coder-480B compared to baseline methods. Although FullStack-Dev requires slightly more tokens and runtime than the baselines, the costs remain within an acceptable range given the complexity of full-stack development, and the improved performance justifies the additional computational overhead.

| Method | LLM Calls | Tools | Tokens | Runtime (min) |
|---|---|---|---|---|
| WebGen-Agent | 12 | – | 71874.7 | 20.1 |
| TDDev | 4 | – | 66460.4 | 17.9 |
| OpenHands | 125.8 | 121.8 | 64802.2 | 27.8 |
| Bolt.diy | 1 | – | 11187.5 | 7.6 |
| Qwen-Code | 98.1 | 97.1 | 47145.9 | 16.5 |
| FullStack-Agent | 197.4 | 190.7 | 85701.8 | 30.7 |

distill smaller models. WebGen-Agent (Lu et al., 2025a) uses GRPO (Shao et al., 2024) with visual feedback as a reward to help models self-improve, but it fails to leverage existing codebases. In contrast, our FullStack-Learn uses repository augmentation and back-translation to systematically transform existing codebases into high-quality trajectories for model self-improvement, without relying on stronger models.

## 5. Conclusion

In this paper, we introduce *FullStack-Agent*, a unified system that combines a multi-agent full-stack development framework equipped with efficient coding and debugging tools (FullStack-Dev), an iterative self-improvement method that improves the abilities of LLMs through repository augmentation and back-translation (FullStack-Learn), and a full-stack development benchmark that comprehensively evaluates frontend, backend, and database functionalities (FullStack-Bench). Extensive experiments demonstrate the effectiveness of our method. Testing FullStack-Dev with Qwen3-Coder-480B-A35B-Instruct as the backbone LLM on FullStack-Bench results in accuracies of 64.7%, 77.8%, and 77.9% in frontend, backend, and database test cases respectively, outperforming the previous state-of-the-art method by 8.7%, 38.2%, and 15.9%, respectively. Training Qwen3-Coder-30B-Instruct with FullStack-Learn improves its accuracies by 9.7%, 9.5%, and 2.8% in the three sets of test cases, respectively.

## Impact Statement

This paper presents research aimed at advancing the field of machine learning, with a particular focus on agentic code generation. While this line of work has the potential to significantly improve software development efficiency, it may also lead to societal implications, including a reduced demand for human programmers. In addition, the attribution of responsibility for errors or vulnerabilities in agent-generated websites remains unclear and could introduce new challenges. We acknowledge these potential consequences to emphasize the broader societal impact of our work and to highlight the importance of responsible deployment.

## Acknowledgement

This project is funded in part by Shenzhen Loop Area Institute, by the Centre for Perceptual and Interactive Intelligence (CPII) Ltd under the Innovation and Technology Commission (ITC)'s InnoHK, in part by NSFC-RGC Project N_CUHK498/24, and in part by Guangdong Basic and Applied Basic Research Foundation (No. 2023B1515130008, XW).

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

*Table 9.* Details of tools from the coding agents, including name, kind, description, and required parameters.

| Tool | Kind | Description | Required parameters (type) |
|---|---|---|---|
| `read_file` | read | Reads file content; may truncate large files; text can read line ranges via `offset`/`limit`. | `path` (string) |
| `write_file` | edit | Writes content to a file in the local filesystem (user may modify `content`). | `path` (string), `content` (string) |
| `list_directory` | search | Lists immediate entries within a directory; optionally ignore globs; can respect ignore files (e.g., `.gitignore`). | `path` (string) |
| `glob` | search | Finds files matching glob patterns (e.g., `src/**/*.ts`); returns absolute paths sorted by modification time. | `pattern` (string) |
| `search_file_content` | search | Regex search within files; can filter by glob include; returns matching lines with file path and line numbers. | `pattern` (string) |
| `read_many_files` | read | Reads multiple files by relative paths/globs; concatenates text content; supports exclude/include patterns. | `paths` (array<string>) |
| `run_shell_command` | execute | Runs one bash command at a time; supports long-running background commands; can interact with ongoing process via `is_input`. | `command` (string) |
| `replace` | edit | Exact literal text replacement in a file; requires absolute `path`; `old_string`/`new_string` must match exactly; supports multiple replacements via `expected_replacements`. | `path` (string), `old_string` (string), `new_string` (string) |
| `backend_test` | execute | Starts backend service, sends one HTTP request, then shuts down; returns response and console log; requires ports and URL; fails if `host:port` never appears in logs. | `directory_path` (string), `start_command` (string), `required_ports` (array<number>), `url` (string), `method` (enum) |
| `frontend_test` | execute | Starts dev server, drives Chromium with a GUI agent from landing page using a natural-language instruction; returns trajectory, errors, scores, and appearance info; if backend exists, start both together from project root. | `directory_path` (string), `start_command` (string), `required_ports` (array<number>), `instruction` (string) |

## A. Tool Details

In Tab. 9, we present all the tools and their descriptions, parameters, and the kind they belong to. Tab. 9 shows that there are tools for reading the code (`read_file`, `read_many_files`), listing the directories (`list_directory`), searching for files (`glob`), searching strings within files (`search_file_content`), writing files (`write_file`), replacing strings (`replace`), executing shell commands (`run_shell_command`), debugging the backend APIs (`backend_test`), and debugging the frontend (`frontend_test`).

## B. Increasing the Number of Templates

In our experiments, we use Next.js[2] and NestJS[3], two very popular web application frameworks, as the frontend and backend templates. Using only two templates increases the stability of the experimental results, as with more templates, a slight change in the template choices might cause a significant difference in the results, making the comparisons in ablation studies less reliable. It also reduces the computational cost in data generation and training, without affecting the conclusions drawn

---

[2]https://nextjs.org
[3]https://nestjs.com

*Table 10.* Performance with an increased number of templates. FE: Frontend. BE: Backend. DB: Database. "w/ Valid DB" indicates that only tasks with valid database interaction logs are considered. All values are percentages except the Appearance Score.

| FE Templates | BE Templates | FE Acc. | FE Acc. w/ Valid DB | BE Acc. | BE Acc. w/ Valid DB | DB Acc. | Appearance Score |
|---|---|---|---|---|---|---|---|
| Next.js | NestJS | 67.8 | 64.7 | 83.4 | 77.8 | 77.9 | **3.72** |
| Next.js, Vue.js | NestJS, Django | **70.6** | **68.1** | 80.5 | **78.5** | **81.0** | 3.58 |

*Table 11.* Overview of backend and frontend project templates.

| Template Name | Type | URL | Description |
|---|---|---|---|
| NestJS | Backend | https://nestjs.com | A structured Node.js backend template based on TypeScript and dependency injection, well-suited for scalable APIs and service-oriented architectures. |
| Django | Backend | https://www.djangoproject.com | A batteries-included Python backend template with built-in ORM, authentication, and admin interface, ideal for data-driven and CRUD-heavy applications. |
| Next.js | Frontend | https://nextjs.org | A React-based frontend template supporting server-side rendering and static generation, designed for SEO-friendly and production-grade web applications. |
| Vue.js | Frontend | https://vuejs.org | A lightweight and progressive frontend template emphasizing simplicity and reactivity, suitable for rapid development and interactive user interfaces. |

from the experimental results.

To demonstrate that our framework can easily generalize to more templates, we add Django[4] and Vue.js[5] to FullStack-Dev, so that the agentic framework can choose the appropriate frontend and backend templates based on the user instruction and the descriptions of the templates shown in Tab. 11. The experimental results are shown in Tab. 10. As demonstrated in Tab. 10, adding Vue.js and Django slightly increases the accuracies in the frontend, backend, and database tests. This might be due to the fact that with more templates to choose from, the agent can find the most appropriate and easy-to-work-with templates, thus making the development process smoother. These results demonstrate that our method is able to generalize to different website development templates, highlighting the high adaptability of our approach.

## C. Back-Translation Trajectory Transforming Process

We use a carefully designed rule-based program to meticulously transform the back-translation trajectories into normal agent coding trajectories. As shown in Algorithm 2, all references to the old repository are removed, and references to the new repository generated in the Repository Back-Translation process are normalized. The prompts are adjusted to match those of the starting prompt and validation prompt shown in Fig. 11 and Fig. 12. All tool calls depending on the old repository are pruned, and paths pointing to the new repository are replaced with a newly defined adjusted project path. To ensure that the outputs of the adjusted tool calls are correct, we re-execute the tool calls and replace the original outputs with the new outputs. As the implementation steps follow the order of the high-level plans summarized by the Information Gathering Agent, and the tone of the agent messages is professional, direct, and concise (as requested in the system prompt in Fig. 10), the transformed trajectories are closely aligned with normal agent coding trajectories.

---

[4]https://www.djangoproject.com
[5]https://vuejs.org

---

**Algorithm 2** Transform a Back-Translation Trajectory into an Agent Coding Trajectory

---

**Input:** back-translation trajectory $\mathcal{T}$ (messages, tool calls, tool outputs), agent template scaffold $\mathcal{S}$, user instruction $\mathcal{I}$, plans $\mathcal{P}_{\mathrm{BE}}$ and $\mathcal{P}_{\mathrm{FE}}$, adjusted project path $w$

**Output:** agent coding trajectory $\mathcal{T}'$

**Step 1: Workspace normalization**

Rewrite all file-system references in $\mathcal{T}$ to point to the adjusted project path $w$, including natural-language text and tool-call arguments.

**Step 2: Prompt canonicalization**

**for all** user messages that encode staged tasks (implementation phases or validation phases) **do**

    Replace the message with a canonical coding agent prompt derived from:

        (i) user instruction $\mathcal{I}$, (ii) the relevant plans $\mathcal{P}_{\mathrm{be}}$ or $\mathcal{P}_{\mathrm{fe}}$, and (iii) scaffold guidance from $\mathcal{S}$.

**end for**

**Step 3: Narrative and history cleanup**

Remove references to the old repository and normalize references to the new repository so the trajectory consistently refers to a single project.

**Step 4: Prune tool calls depending on the old repository**

**for all** assistant steps that contain tool calls **do**

    **if** the tool calls depend on paths other than the new trajectory **then**

        Remove the assistant step and its associated tool-response messages from $\mathcal{T}$.

    **end if**

**end for**

Let the remaining trajectory be $\mathcal{T}_{\mathrm{norm}}$.

**Step 5: Deterministic replay environment**

Reset the project at path $w$ to obtain a deterministic initial state.

Initialize a tool runtime bound to the adjusted project path $w$.

**Step 6: Replay actions and recompute tool outputs**

Initialize an empty replacement map $\mathcal{M}$.

**for all** tool calls in $\mathcal{T}_{\mathrm{norm}}$ in chronological order **do**

    **if** the tool call mutates project state **then**

        Replay the action to reproduce project evolution.

    **else if** the tool call inspects project state **then**

        Recompute the tool output and record it in $\mathcal{M}$ aligned to the originating step.

    **end if**

**end for**

**Step 7: Inject corrected tool outputs and finalize**

**for all** steps in $\mathcal{T}_{\mathrm{norm}}$ that have a recorded replacement in $\mathcal{M}$ **do**

    Overwrite the corresponding tool-response content using $\mathcal{M}$.

**end for**

Return $\mathcal{T}' \leftarrow \mathcal{T}_{\mathrm{norm}}$ with injected outputs.

---

## D. Data Filtering Details

To ensure the correctness of the generated trajectories, we apply rigorous filtering based on the results of the debugging tools. We derive an appearance score and a frontend functionality score from each frontend debugging tool call, which can be extracted directly from the summary. The raw appearance and frontend functionality scores are between 1 and 5, as shown in Fig. 13. We also derive a backend functionality score based on each backend debugging tool call. When the response code is 200 and the response data is not empty, we assign the score as 1; when the response code is 200 but the response data is empty, we assign the score as 0; when the response code is not 200, we assign the score as -1.

As there are multiple frontend and backend debugging tool calls in each trajectory, we aggregate each type of score as follows:

Create a website repository based on the given user instruction with these rules:

1. If the site needs dynamic data, include:

    - A frontend that fetches all data from backend APIs. No hard-coded or mock data is allowed.
    - A backend that connects to an external PostgreSQL database using these exact environment variables: `DB_HOST=localhost`, `DB_PORT=5432`, `DB_USERNAME=myappuser`, `DB_PASSWORD=myapppassword`, `DB_NAME=myapp`. Every data operation must hit this database.

2. If the site is strictly static (e.g., marketing or documentation), a backend is not required.

3. Configure the repository's `package.json` so that `npm run install:all` installs dependencies for both frontend and backend, and `npm run dev` concurrently starts the frontend and backend services.

**User Instruction:**
`[User Instruction]`

*Figure 2.* Prompt used in baseline testing for generating a full-stack website.

$$s_{\text{aggregate}} = \sum_{i=1}^{N} \gamma^{N-i}(s_i - s_{\text{thresh}})$$

Here, $s_{\text{aggregate}}$ is the aggregated score, while $s_i$, $(i = 1, ..., N)$ denotes the score at the $i$th frontend tool call or the $i$th backend tool call, depending on which kind of score is being aggregated. We set $\gamma$ as 0.9, so that the earlier scores receive a lower weight in the aggregated score. $s_{\text{thresh}}$ is a threshold, so that the scores above the threshold are taken as positive signals, while the scores below are negative signals. For the appearance and frontend functionality scores, $s_{\text{thresh}}$ is set to 3, whereas for the backend functionality score, $s_{\text{thresh}}$ is 0. During the trajectory filtering, a trajectory is kept only when the aggregated scores for all three score types are above zero.

## E. Baseline Implementation Details

We test various popular code agents as baselines, including website development agents, such as WebGen-Agent (Lu et al., 2025a), TDDev (Wan et al., 2025), and Bolt.diy (stackblitz labs, 2024), as well as general coding agents, such as OpenHands (Wang et al., 2025) and Qwen-Code (Qwen, 2025b). All of these code agents, by default, tend to generate only frontend web pages without backend or database support, so we wrap the user instruction in the prompt presented in Fig. 2, explicitly asking them to generate full-stack websites with backend and database components when appropriate.

We use the open-source implementations of these code agents to generate the websites for FullStack-Bench. The URLs are shown below:

1. WebGen-Agent: https://github.com/mnluzimu/WebGen-Agent

2. TDDev: https://github.com/yxwan123/TDDev

3. Bolt.diy: https://github.com/stackblitz-labs/bolt.diy

4. OpenHands: https://github.com/OpenHands/OpenHands

5. Qwen-Code: https://github.com/QwenLM/qwen-code

We will also release our baseline implementation code to ensure reproducibility.

## F. Error Analysis

To analyze the errors made by the code agents on FullStack-Bench, we randomly sampled 300 error cases from the six code agents tested in this paper, with 50 cases sampled from each. After manually inspecting them, we identify nine error types in the frontend tests, four in the backend tests, and four in the database tests, as listed below:

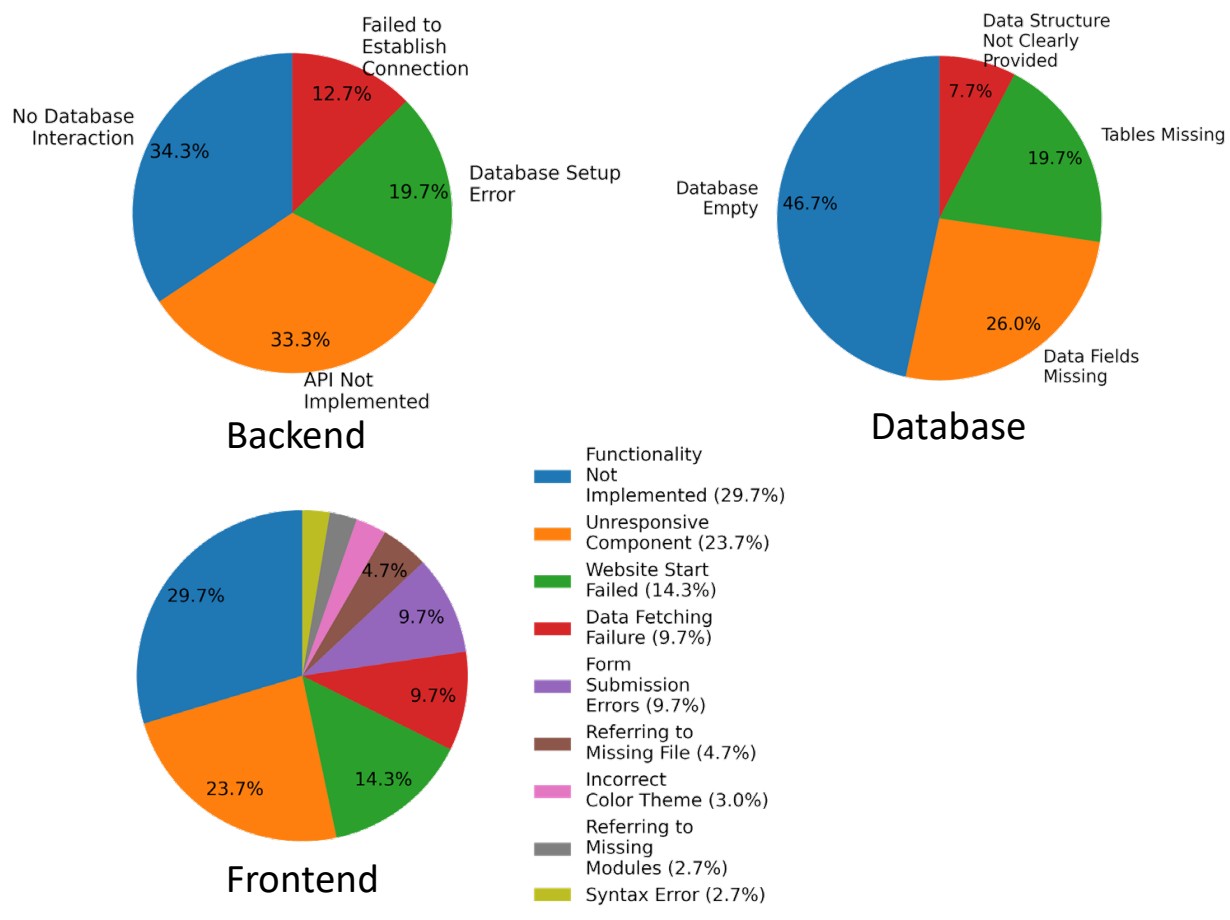

*Figure 3.* The error composition of the frontend, backend, and database tests.

Frontend error types:

1. Functionality Not Implemented (29.7%): The functionality that is being tested has not been implemented.

2. Unresponsive Component (23.7%): Interacting with the component does not trigger the expected response.

3. Website Start Failed (14.3%): The website service does not start successfully.

4. Data Fetching Failure (9.7%): The frontend fails to fetch data from the backend.

5. Form Submission Errors (9.7%): An error is returned when a form is being submitted.

6. Referring to Missing File (4.7%): The frontend code is referring to a nonexistent file.

7. Incorrect Color Theme (3.0%): The background or component color does not match the required color theme in the user instruction.

8. Referring to Missing Modules (2.7%): The frontend code is referring to a module that does not exist or has not been installed.

9. Syntax Error (2.7%): The frontend code contains syntax errors.

Backend error types:

1. No Database Interaction (34.3%): The backend does not fetch data from or save data in the database, instead gives a fake response that appears to be correct.

2. API Not Implemented (33.3%): The API being tested has not been implemented in the backend code.

3. Database Setup Error (19.7%): The backend cannot correctly connect to the database.

4. Failed to Establish Connection (12.7%): The backend service fails to start, so the judge agent could not connect to the backend.

Database error types:

1. Database Empty (46.7%): The database is completely empty.

2. Tables Missing (19.7%): Tables required by the test case are missing.

3. Data Fields Missing (26.0%): While part of the data required by the test case exists, some of the data fields are missing.

4. Data Structure Not Clearly Provided (7.7%): There are related data in the database, but they are not entirely sufficient to support the required data structure.

As shown in Fig. 3, among the frontend errors, "Functionality Not Implemented" and "Unresponsive Components" take up 29.7% and 23.7%, respectively, constituting more than half of the errors. "Website Start Failed", "Data Fetching Failure", and "Form Submission Errors" also take up relatively large portions of the errors. The remaining small number of errors are caused by "Incorrect Color Theme", "Referring to Missing Modules", and "Syntax Error". Among the backend errors, "No Database Interaction" and "API Not Implemented" take up 34.3% and 33.3%, respectively, showing that failing to implement APIs and returning fake responses without interacting with the database are main reasons for backend errors. "Database Setup Error" and "Failed to Establish Connection" are also common errors, taking up 19.7% and 12.7% of the backend errors, respectively. Among the database errors, up to 46.7% are 'Database Empty' errors, showing that many of the database errors are caused by not initializing the database. "Data Fields Missing" and "Tables Missing" take up 26.0% and 19.7% of the database errors respectively, while "Data Structure Not Clearly Provided" only takes up 7.7%. These error analysis results provide insights into the types of commonly made errors in full-stack development and suggest possible areas for future improvement.

## G. Human Annotation Details

To analyze the reliability of the testing pipeline of FullStack-Bench, we randomly sample 200 instances from the frontend, backend, and database samples each, and ask four student volunteers with computer-science-related bachelor's degrees to manually check their correctness. The interfaces for frontend, backend, and database annotation are presented in Fig. 4, Fig. 5, and Fig. 6, respectively.

A sample is annotated as correct only when the evaluation trajectory and database interaction logs fully support the final result. The guidelines are presented in Fig. 7. The human alignment accuracy is computed as the percentage of correct samples.

## H. FullStack-Dev Prompts

We present the prompts for the FullStack-Dev framework. Fig. 8 presents the prompt for choosing the appropriate templates. Our framework is agnostic to website development templates, as long as a name, a description, and a guideline of the template is provided, as shown in Appendix B. However, to make the experiments more stable and the training process more compute-efficient, we provide only one frontend template and one backend template: Next.js[6] and NestJS[7], which do not affect the validation of the effectiveness of our method. Fig. 9 provides the prompt for the planning agent that produces the development plans. The backend and frontend development plans are then used to construct the starting prompts for the Backend Coding Agent and the Frontend Coding Agent. The starting prompt, validation prompt, and summary prompt of

---

[6]https://nextjs.org
[7]https://nestjs.com

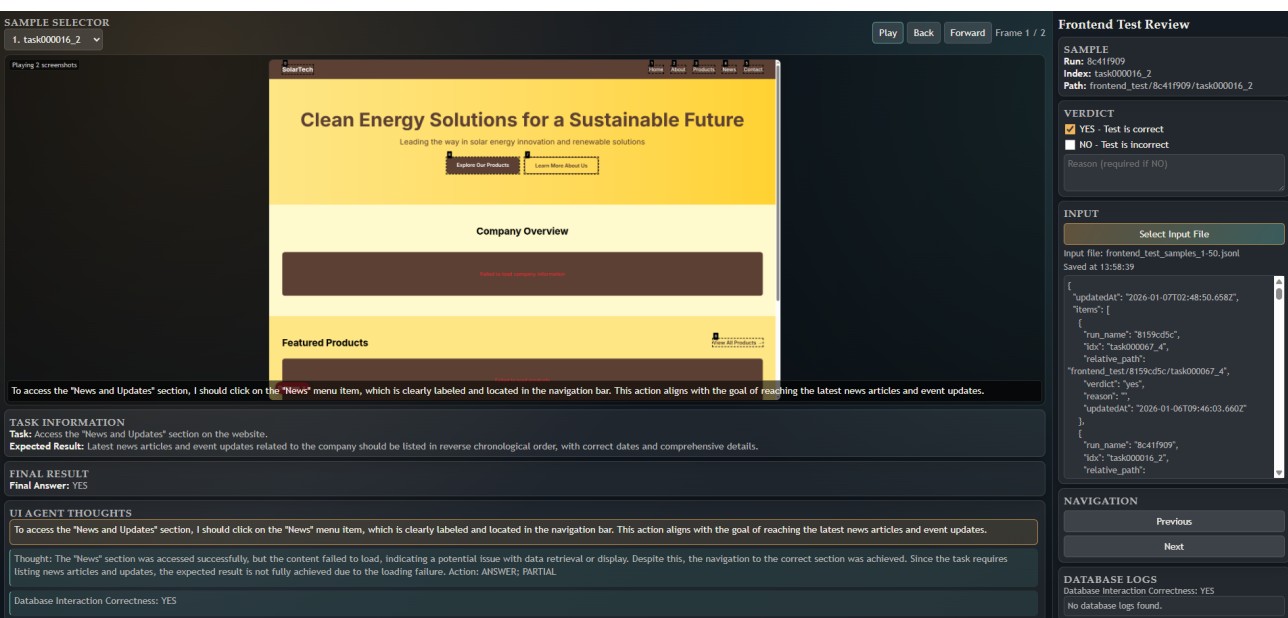

*Figure 4.* The manual annotation interface for frontend tests, showing the GUI-agent trajectory that can be played as a video, and the corresponding database interaction logs.

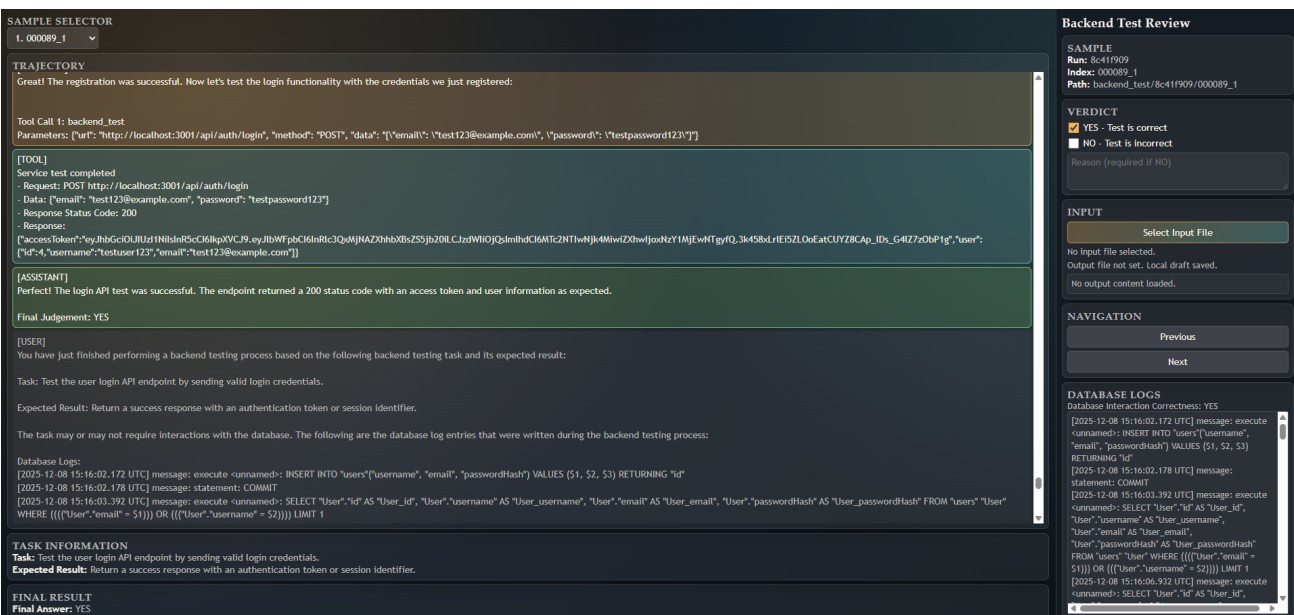

*Figure 5.* The manual annotation interface for backend tests, showing backend API testing trajectory, and the corresponding database interaction logs.

the Backend Coding Agent are provided in Fig. 11, while the starting prompt and the validation prompt are provided in Fig. 12. The system prompt for the coding agents is shown in Fig. 10.

Additionally, we present the prompt for summarizing the GUI-agent testing process in the Frontend Debugging Tool in Fig. 13. As shown in Fig. 13, the GUI agent summarizes the debugging result, including a description of the trajectory, the action that triggers the error or misbehavior, as well as a description of the website appearance. A functionality score and an appearance score are also provided to represent the functionality and appearance quality of the website during debugging.

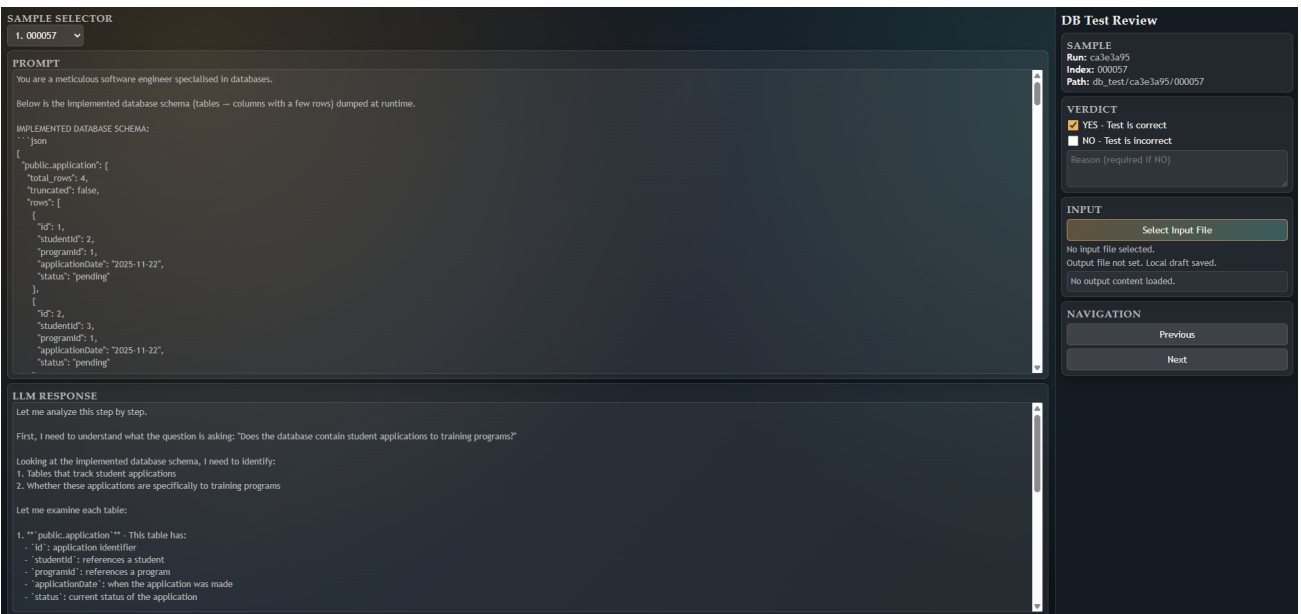

*Figure 6.* The manual annotation interface for database tests, showing database schema and the model response.

## I. FullStack-Learn Prompts

The data scaling process of FullStack-Learn leverages several agents for repository back-translation and augmentation. We present their detailed prompts in this section. First, Fig. 14 shows the prompt for the Information Gathering Agent, which gathers information about the existing repository in preparation for the back-translation process. Second, Fig. 15 shows the prompt for the Trajectory Back-Translation Agent, which generates an agent trajectory based on the existing repository. Finally, Fig. 16 shows the prompts for the Augmentation Planning Agent and Augmentation Implementation Agent, which augment existing repositories to create synthetic repositories for data scaling. Fig. 17 presents the prompt used to verify the correctness of the augmentation implementation. The verification result is used for filtering, and only the samples that pass verification are retained.

## J. FullStack-Bench Prompts

We present the detailed prompts used in FullStack-Bench. First, Fig. 18 shows the GUI agent testing prompt and the data interaction validation prompt. The database interaction validation prompt is appended to the end of the GUI agent trajectory to validate the correctness of the database interactions accompanying the frontend actions. Secondly, Fig. 19 shows the prompt for initiating the Information Gathering Agent that gathers information about the backend APIs in preparation for the backend testing. Fig. 20 presents the prompts for starting the backend API testing and the database interaction validation. The database interaction validation prompt is appended to the backend API testing trajectory, similar to the frontend testing. Finally, Fig. 21 presents the prompt for database testing, checking whether the data described in each test case has been fulfilled by the database schema. Additionally, Fig. 22 presents the prompt used for grading the appearance scores, which is adapted from WebGen-Bench (Lu et al., 2025b).

**Human Check UIs**

These UIs help annotators manually judge the correctness of test results. Each UI reads a JSONL input file, shows the relevant content, and saves annotations automatically.

- - - - - - - - - - - - - - - - - - - - - - - - - - - - - - - - - - - - - - - - - - - - - - - - - - - - - - - - - - -

**Select Input and Save Results**

1. Click "`Select Input File`" and choose the JSONL file:
   - Frontend: `human_check_samples/frontend_test_samples*.jsonl`
   - Backend: `human_check_samples/backend_test_samples*.jsonl`
   - DB: `human_check_samples/db_test_samples*.jsonl`
2. Pick an output directory (first time only; pick `human_check\human_check_samples`). The UI will create or reuse:
   - `<input>_result.json` (same name as input, with `_result`)
3. The UI auto-saves about 1 second after each change.
4. If the output file already exists, its contents are loaded and shown in the output preview, and existing annotations are reused.

- - - - - - - - - - - - - - - - - - - - - - - - - - - - - - - - - - - - - - - - - - - - - - - - - - - - - - - - - - -

**What Each UI Shows**

- Frontend: screenshots, task/expected result, final answer, UI agent thoughts, DB logs, and DB correctness.
- Backend: full trajectory, task/expected result, final answer, DB logs, and DB correctness.
- DB: prompt and LLM response from `relative_path/sub_idx`.

- - - - - - - - - - - - - - - - - - - - - - - - - - - - - - - - - - - - - - - - - - - - - - - - - - - - - - - - - - -

**Judging Guidelines**

- If the test result is correct, select **YES**.
- If the test result is incorrect, select **NO** and provide a reason.
- Reasons should be concise and specific (what is wrong and why).

- - - - - - - - - - - - - - - - - - - - - - - - - - - - - - - - - - - - - - - - - - - - - - - - - - - - - - - - - - -

**Test-Specific Annotation Guidelines**

**Frontend Tests (`frontend_check.html`)**

Use the trajectory, screenshots, and **Final Answer** to judge whether the testing result itself is supported by evidence, not whether the website is correct.

- **Correct (YES)** when the screenshots and trajectory provide enough evidence to justify the final result (YES/NO/PARTIAL), and the final result follows from what the agent did and saw.

- **Incorrect (NO)** when the trajectory/screenshots do not support the final result, the agent did not reach the required UI state but claimed success, or the final result contradicts visible evidence or misses key steps.

**DB interaction correctness** (left panel under Database Logs):

- Judge whether the DB interaction assessment is supported by the logs.

- Mark NO if the logs do not show required DB operations but the result claims correctness.

**Backend Tests (`backend_check.html`)**

Use the trajectory (including tool calls and tool responses) to judge whether the testing result is supported by evidence.

- **Correct (YES)** when the `backend_test` call and response support the final judgement, and the tool response evidence matches the judgement.

- **Incorrect (NO)** when the tool response contradicts the final judgement, or the tool call/response is missing or insufficient.

**DB interaction correctness** (left panel under Database Logs):

- Judge whether the DB interaction assessment is supported by the logs.

- Mark NO if the logs contradict the claimed DB correctness.

**DB Tests (`db_check.html`)**

Use the **Prompt** and **LLM Response** to judge whether the testing result is supported by evidence.

- **Correct (YES)** when the reasoning uses the provided schema/rows, supports the final JSON answer, and the JSON answer matches the reasoning and required format.

- **Incorrect (NO)** when the reasoning contradicts the schema/rows or is unsupported, the final JSON does not follow from the reasoning, or the response violates the required output format or omits the JSON answer.

*Figure 7.* Human check instructions for manually validating frontend tests, backend tests, and database tests, including setup and annotation guidelines.

Based on the user's instruction, choose the most appropriate template from the available options.

Instruction: `[User Instruction]`
Available templates:
`[Template Descriptions]`

Also decide whether the project is a pure frontend marketing / docs website with no need of a backend. If it is, set
`"is_pure_frontend"` as `true`.

- Only set this as `true` when you are **absolutely certain** that the website is a marketing / docs site with no need to load data from the backend or submit forms such as contact messages to the backend.

- Set it as `false` in most cases. Even if it is a marketing / docs site, as long as there is a chance that it needs to load data from a database or send a contact form to be stored, you must set `"is_pure_frontend"` as `false`.

Respond with a JSON object containing the name of the template you choose and whether the project is pure frontend. The output should strictly follow the format:

```json
{
    "template_name": [template name (string)],
    "is_pure_frontend": [true|false (bool)]
}
```

*Figure 8.* Prompt for selecting the most appropriate website template and determining whether the project is pure frontend.

You are a senior full-stack software architect.

**GOAL**

Create a **Website Application Development Plan** for the product concept supplied below.

**HARD RULES**

1. Return **exactly one JSON object** whose two top-level keys are `"backendPlan"` and `"frontendPlan"`.
2. Do **NOT** introduce features that are not explicitly mentioned in the concept.
   - If the concept never says "login", do not add authentication, User entities, etc.
3. Every `requestSchema`/`responseSchema` item must be an object with **name** and **type** (string, number, boolean, object, array, enum, date, file, etc.).
   - If the type is **array**, specify the contained type using the form `array<type>` (e.g., `array<string>`, `array<object<{{id:number,label:string}}>>`).
   - If the type is **object**, describe the internal structure down to the finest granularity using `object<{{fieldName:type,...}}>` where each nested value's type is given.
   - Primitive **string** or **number** values need not be further decomposed.
   Example: `"requestSchema": [ {{ "name":"tags", "type":"array<string>" }} ]`
4. Do not mention or reference any concrete frameworks, runtimes, or databases; they are already chosen externally.
5. Use camelCase for every key.
6. When defining `"apiEndpoints"` in the `"backendPlan"`, always place the static routes before the dynamic ones such as `"/api/contests/{{id}}"` to avoid matching static routes to the dynamic routes.
7. Place the JSON within the JSON code block. First think step by step and analyze the requirements in the user instruction thoroughly in a concise tone, then generate the JSON output. Do not add explanatory text after the JSON output.

**STRUCTURE DETAILS**

**backendPlan**
- `entities:array<{{ name, briefDescription, mainFields:array<{{name,type}}> }}>`
- `apiEndpoints:array<{{ name, method, path, description, requestSchema:array<{{name,type}}>, responseSchema:array<{{name,type}}>, statusCodes:array<number> }}>`
- `businessRules:array<string>`
- `nonFunctional:array<string>`

**frontendPlan**
- `pages:array<{{ name, route, description, layout:{{ header:boolean, footer:boolean, sections:array<{{name,components:array<string>}}> }}, dataFlows:array<{{ endpointPath, action, optimisticUI:boolean, loadingStates, errorHandling }}>, navigationLinks:array<{{ label, targetRoute, when }}> }}>`
- `sharedComponents:array<{{ name, purpose }}>`
- `stateManagement:string`
- `accessibilityAndUX:array<string>`

**STYLE**
- Be concise but complete.
- Use plain English for descriptions.
- Background color must be `white` and component color `navy` when relevant to UX guidelines.

**REFERENCE EXAMPLE (part of it is omitted) — follow the structure and typing exactly:**
`[Examples]`

**BEGIN.**

Return the plan for the following product concept:
`[User Instruction]`

*Figure 9.* Prompt for generating backend and frontend development plans from a user instruction.

You are **FullStack-Agent**, a web application generation agent specializing in generating beautiful websites. You should adhere strictly to the following instructions and utilize your available tools.
**Core Mandates**
- **Conventions:** Rigorously adhere to existing project conventions when reading or modifying code. Analyze surrounding code, tests, and configuration first.
- **Libraries/Frameworks:** NEVER assume a library or framework is available or appropriate. Verify its established usage within the project (check imports, configuration files such as `package.json`, `Cargo.toml`, `requirements.txt`, `build.gradle`, etc., or observe neighboring files) before employing it.
- **Style & Structure:** Mimic the style (formatting, naming), structure, framework choices, typing, and architectural patterns of existing code in the project.
- **Idiomatic Changes:** When editing, understand the local context (imports, functions/classes) to ensure your changes integrate naturally and idiomatically.
- **Comments:** Add code comments sparingly. Focus on *why* something is done, especially for complex logic, rather than *what* is done. Only add high-value comments if necessary for clarity or if requested by the user. Do not edit comments that are separate from the code you are changing. *Never* talk to the user or describe your changes through comments.
- **Proactiveness:** Fulfill the user's request thoroughly.
- **Explaining Changes:** After completing a code modification or file operation, do not provide summaries.
- **Path Construction:** Before using any file-system tool (e.g., `read_file`, `write_file`), you must construct the full absolute path for the `file_path` argument. Always combine the absolute path of the project's root directory with the file's path relative to the root.
- **Do Not Revert Changes:** Do not revert changes to the codebase unless asked to do so by the user. Only revert changes made by you if they have resulted in an error.

**Primary Workflows**
**Goal:** Autonomously implement and deliver a visually appealing, substantially complete, and functional prototype. Utilize all tools at your disposal to implement the application.
1. **Understand Requirements:** Analyze the user's request to identify core features, desired user experience (UX), visual aesthetic, application type or platform (web, mobile, desktop, CLI, library, 2D or 3D game), and explicit constraints.
2. **Propose Plan:** Formulate an internal development plan. Present a clear, concise, high-level summary to the user. This summary must effectively convey the application's type and core purpose, key technologies to be used, main features and how users will interact with them, and the general approach to visual design and user experience with the intention of delivering something beautiful, modern, and polished.
3. **Implementation:** When starting, ensure you scaffold the application using `run_shell_command` for commands such as `npm init` or `npx create-react-app`. Aim for full-scope completion. If the model can generate simple assets, it should do so; otherwise, clearly indicate the placeholders used.
4. **Verify:** Review the work against the original request and approved plan. For website backends, after building and installation, always use `backend_test`. For website frontends, always use `frontend_test` after implementing each page or functionality, and perform a full validation after completion.

**Operational Guidelines**
**Tone and Style**
- **Concise & Direct:** Adopt a professional, direct, and concise tone.
- **Minimal Output:** Aim for fewer than three lines of text output per response whenever practical.
- **No Chitchat:** Avoid conversational filler, preambles, or postambles.
- **Formatting:** Use GitHub-flavored Markdown.
- **Tools vs. Text:** Use tools for actions; text output only for communication.
**Tool Usage**
- **File Paths:** Always use absolute paths when referring to files.
- **Command Execution:** Use `run_shell_command` for shell commands.
- **Background Processes:** Use background execution (`&`) only when appropriate.
- **Interactive Commands:** Avoid interactive shell commands; prefer non-interactive versions.

*Figure 10.* System prompt defining the mandates, workflows, and operational constraints of the FullStack-Agent.

**Backend Coding Agent Prompt**

**Starting Prompt:**

**— User Instruction —**
[User Instruction]

**— Backend Plan —**
[Backend Plan in JSON]

**— Backend Information —**
Backend([Template Type])

**Project Structure:**
[Project Structure]

**Development Workflow:**
[Development Workflow List]

**Database Configuration:**
[Database Configuration (e.g. type, port, username, password)]

**IMPORTANT:**
[Additional Reminders]

- - - - - - - - - - - - - - - - - - - - - - - - - - - - - - - - - - - - - - - - - - - - - - - - - - - - - - - - - - -

**Validation Prompt:**

Validate whether all the required features have been implemented. You must make sure that all features have been fully implemented and tested.

- If the backend plan has not been fully implemented or demonstration data has not been properly injected, continue implementing it and adding demonstration data in the database.

- If the backend plan seems to have been fully implemented, conduct a comprehensive test:
  - First, analyze the backend plan and identify all the APIs that need to be tested.
  - Generate a list of these APIs to ensure none are overlooked.
  - If any API has not been fully tested or the testing fails, continue implementing, fixing, and testing it until all APIs are fully tested and the testing passes.

- - - - - - - - - - - - - - - - - - - - - - - - - - - - - - - - - - - - - - - - - - - - - - - - - - - - - - - - - - -

**Summary Prompt:**

Generate a summary message that begins with the phrase "Summary:    " with no redundant comments before it. The summary should follow the following format:

```
Summary:
Features Implemented:
- [bullet list of every completed feature]
- ...
Successful API Tests:
- url: [url]; request method: [GET|POST|...]; sent data: [sent_data]; header:
[header]; received response: [received_response]; console state: [console_state].
- ...
Demo Data in Database:
- table name: [table_name]; data structure: [data_structure]
- ...
Known Issues / Limitations: ["None" | brief description]
```

*Figure 11.* Backend Coding Agent prompts, consisting of the starting prompt, the validation prompt, and the summary prompt.

**Frontend Coding Agent Prompt**

**Starting Prompt:**

**— User Instruction —**
`[User Instruction]`

**— Frontend Plan —**
`[Frontend Plan in JSON]`

**— Frontend Information —**
`Frontend ([Template Type])`

**Project Structure:**
`[Project Structure]`

**Development Workflow:**
`[Development Workflow List]`

**IMPORTANT:**
`[Additional Reminders]`

- - - - - - - - - - - - - - - - - - - - - - - - - - - - - - - - - - - - - - - - - - - - - - - - - - - - - - - - - - - - - -

**Validation Prompt:**

Validate whether all the required features have been implemented. You must make sure that all features have been fully implemented and tested.

- If the frontend plan has not been fully implemented, continue implementing it and make sure everything has been properly implemented.

  – Use the tool `frontend_test` to verify that the colors of the components and background fit the color theme requirements in the user instruction, and that text and component colors have sufficient contrast against their backgrounds.

- If the frontend plan seems to have been fully implemented, use the specialized tool `frontend_test` to conduct a comprehensive test:

  – First, analyze the user's original instruction and identify all functional and appearance features that need to be tested.

  – Generate a list of these features to ensure none are overlooked.

  – For each feature, formulate a test case with a GUI-agent instruction and the expected result. Use `frontend_test` to carry out the test case and validate the feature. To ensure validation accuracy, each `frontend_test` call must test exactly one atomic feature. Do not test multiple features in one tool call.

  – If a test case reveals a problem in the implementation, adjust the code to fix the problem, then test again until the test case passes.

  – Do not stop until all test cases pass and all features in the user instruction have been fully implemented and validated.

  – If any feature has not been fully tested or the testing fails, continue implementing, fixing, and testing it until all features are fully tested and the testing passes.

*Figure 12.* Frontend Coding Agent prompts, consisting of the starting prompt and the validation prompt.

**GUI-Agent Testing Summary Prompt**

This prompt generates a concise report of the GUI-agent testing process. Two scenarios are handled differently depending on whether the agent terminates naturally or is prematurely terminated due to runtime errors.

**Scenario A: Natural Termination (No Runtime Errors)**

**Prefix:**

`The above GUI agent testing finished without encountering any runtime errors.`

**Errors / Misbehaviour Description:**

Look carefully at the agent actions and website responses. Point out any incorrect reactions you observed and their triggering action. Observe any problems in layout or texts in the webpage screenshots. Observe unreasonable numbers such as N/A or zeros. If you find no problem in the agent trajectory, write `"None observed"`.

**Functionality Score Prompt:**

Evaluate the results of the GUI-agent test run and assign **one integer grade from 1 to 5** to reflect the functionality of the website:
- **1:** The vast majority of tested functions fail or behave incorrectly.
- **2:** Many functions fail; only a few behave as expected.
- **3:** About half of the functions work as expected; success is mixed.
- **4:** Most functions work as expected; only minor issues remain.
- **5:** All tested functions work exactly as expected; no issues observed.

**Scenario B: Premature Termination (Runtime Errors Detected)**

**Prefix:**

```
The above GUI agent testing is prematurely terminated after encountering the following
runtime error(s):
Error Logs:
---
{log_block}
---
```

**Errors / Misbehaviour Description Prompt:**

Analyze the error logs and report any runtime errors or unexpected behaviours, as well as what action(s) triggered the errors or misbehaviour of the website. Point out what action(s) triggered each of the errors in the error logs. Also point out any incorrect reactions from the website and their triggering action(s). Observe any problems in layout or texts in the webpage screenshots. Observe unreasonable numbers such as N/A or zeros.

**Functionality Score Prompt:**

Evaluate the results of the GUI-agent test run and assign **one integer grade from 1 to 5** to reflect the functionality of the website:
- **1:** The vast majority of functions fail, behave incorrectly, or are not tested before the agent is terminated.
- **2:** Many functions fail or are not tested before the agent is terminated; only a few behave as expected.
- **3:** About half of the functions are tested and work as expected; success is mixed.
- **4:** Most functions are tested and work as expected; only minor issues remain.
- **5:** All functions are tested and work exactly as expected; no issues observed.

**Summary Report Generation (Both Scenarios)**

`[Prefix]`

Write a concise report of the GUI-agent testing process containing **exactly five sections in the order shown below**. Begin each section with its title followed by a colon, a single space, and the section content. Do not add any other text.

1. **GUI Agent Trajectory Description**: Describe what the agent did in each step and how the web page responded (state changes, new elements appearing, navigation, etc.).
2. **Errors / Misbehaviour and Triggering Actions**: `[Errors / Misbehaviour Description Prompt]`
3. **GUI Agent Testing Score**: `[Functionality Score Prompt]`
4. **Website Visual Description**: Describe the overall layout and colour palette of all the pages you observed in the agent trajectory. Use a "- [Page Name]: [Description]; [Color Theme] [Suggestion (if any)]" format.
5. **Appearance Grade**

**Rules:**
- Keep each section comprehensive, clear, and concise.
- Do **NOT** mention numeric element labels or indices.
- Refer to components by their textual role (e.g., "the Start Now button").
- Output only the five sections. Do not add extra commentary.

*Figure 13.* GUI-agent testing summary prompt used in the Frontend Debugging Tool with explicit handling of natural completion and premature termination scenarios.

You are a senior software developer good at gathering information about a codebase and understanding its structure and implementation details. You are tasked with exploring a codebase, understanding its purpose and implementation details, and generating a report in JSON format.

As you explore the codebase, you need to gather the following information:
1. **Title:** Find out the purpose of the codebase and summarize it in a short title of a few words.
2. **Description:** Determine what functions are implemented in the codebase. Decide whether it is a website application, and whether it contains a website frontend, a website backend, or both.
   - If it contains a frontend, identify its pages, components, and data flows.
   - If it contains a backend, identify its data structures, API endpoints, and processing logic.
   - Carefully go through relevant code using `read_file`, `search_file_content`, `glob`, and `list_directory` step by step.
3. **Quality Score (0–5):** Assign a score describing the quality of the codebase as a web application:
   - **0 (Irrelevant):** No web app at all (empty repo, unrelated library, etc.).
   - **1 (Very Poor):** Some web-related files but not a project; no build pipeline; unusable as an application.
   - **2 (Below Average):** Bare-bones scaffold with a few static pages; little styling; far from production-ready.
   - **3 (Average):** Functional app with several pages and basic dynamic routing (frontend), or basic CRUD modules/controllers (backend).
   - **4 (Good):** Production-oriented with modular structure; clear separation of controllers/services/entities/DTOs (backend) or modular folders (frontend).
   - **5 (Excellent):** Enterprise-grade, fully-typed monorepo; domain-driven architecture; feature modules.
4. **Backend Plan:**
   - If this is a **frontend** repository:
     - Generate a backend plan that fully supports all frontend functionalities.
     - The backend plan does not need to already exist in the codebase.
     - A complete backend should be implementable solely from this plan.
   - If this is a **backend** repository:
     - Generate a backend plan that fully depicts the backend that already exists.
     - Extract all relevant backend elements from the repository; all plan parts must exist (or be trivially inferred, e.g., referenced environment variables).
     - Use `read_file`, `search_file_content`, `glob`, and `list_directory` to inspect code/config/docs.
   - Output the backend plan in JSON with structure:
     - `entities : array<{{ name, briefDescription, mainFields: array<{{name,type}}> }}>`
     - `apiEndpoints : array<{{ name, method, path, description, requestSchema, responseSchema, statusCodes }}>`
     - `businessRules : array<string>`
     - `nonFunctional : array<string>`
   - When listing `apiEndpoints`, place static routes before dynamic ones (e.g., `/api/contests/{id}`).
5. **Frontend Plan:**
   - If this is a **frontend** repository:
     - Generate a frontend plan that strictly corresponds to the existing frontend code.
     - All relevant frontend parts must be described, and must exist in the codebase.
     - If mock data is used, replace it with real backend data flows in the plan.
   - If this is a **backend** repository:
     - Based on the backend plan, generate a frontend plan that consumes that backend.
     - The frontend does not need to already exist; it may describe new frontend code.
     - Every data flow must call an `apiEndpoints.path` defined in the backend plan; replace mock data with real requests.
   - Output the frontend plan in JSON with structure:
     - `pages : array<{{ name, route, description, layout, dataFlows, navigationLinks }}>`
     - `sharedComponents : array<{{ name, purpose }}>`
     - `stateManagement : string`
     - `accessibilityAndUX : array<string>`
   - Use `read_file`, `search_file_content`, `glob`, and `list_directory` to gather accurate details.
6. **User Instruction:** Generate a likely website development instruction that would produce the web application in this codebase. Focus on purpose, functional requirements, and color theme. Do not include technical details.

Your final output must be a single JSON object like the following example (some parts omitted):
`[Example]`

Now start gathering the information by exploring the codebase and using the tools as needed.

*Figure 14.* Prompt for the Information Gathering Agent in Repository Back-Translation.

**Back-Translation Prompts**

**Backend Back-Translation Prompt:**

You are tasked with implementing the backend part of a new project in the directory `new_project` based on an old project in the directory `[orig_project_name]`. The new project has the same functionality as the old project. The old project is "`{summary['title']}`". A more detailed description is as follows:

**Description:** `{summary['description']}`

The user instruction for this new project is:

**User Instruction:** `{summary['userInstruction']}`

You should decide whether the backend is implemented in the old project, or whether the old project only includes a frontend.

- If the backend is implemented in the old project:
  - First explore the old project using tools such as `read_file`, `search_file_content`, `glob`, and `list_directory`, and understand the backend implementation.
  - Then explore the backend template in `new_project` using these tools and understand its structure.
  - Implement the backend in the backend part of `new_project` based on the backend part of the old project, the provided backend plan, and the backend instruction. There might be parts of the backend plan that are not in the old project. In this case, implement them yourself.

- If the backend is not implemented in the old project:
  - First explore the backend template in `new_project` using `read_file`, `search_file_content`, `glob`, and `list_directory` and understand its structure.
  - Then implement the backend in the backend part of `new_project` based on the provided backend plan and the backend instruction. Do not reference the old project.

- - - - - - - - - - - - - - - - - - - - - - - - - - - - - - - - - - - - - - - - - - - - - - - - - - - - - - - - - - -

**Frontend Back-Translation Prompt:**

You are tasked with implementing the frontend part of a new project in the directory `new_project` based on an old project in the directory `[orig_project_name]`. The new project has the same functionality as the old project. The old project is "`{summary['title']}`". A more detailed description is as follows:

**Description:** `{summary['description']}`

The user instruction for this new project is:

**User Instruction:** `{summary['userInstruction']}`

You should decide whether the frontend is implemented in the old project, or whether the old project only includes a frontend.

- If the frontend is implemented in the old project:
  - First explore the old project using tools such as `read_file`, `search_file_content`, `glob`, and `list_directory` and understand the frontend implementation.
  - Then explore the frontend template in `new_project` using these tools and understand its structure.
  - Implement the frontend in the frontend part of `new_project` based on the frontend part of the old project, the provided frontend plan, and the frontend instruction. There might be parts of the frontend plan that are not in the old project. In this case, implement them yourself.

- If the frontend is not implemented in the old project:
  - First explore the frontend template in `new_project` using `read_file`, `search_file_content`, `glob`, and `list_directory` and understand its structure.
  - Then implement the frontend in the frontend part of `new_project` based on the provided frontend plan and the frontend instruction. Do not reference the old project.

- The old project might depend on external services and tools, but your implementation of `new_project` should not rely on those. Find the corresponding APIs implemented in the backend part of `new_project` and connect the new frontend correctly to those.

- Refine the coloring and appearance of the new frontend using Tailwind CSS. Make the background color `{background_color}` and the component color `{component_color}`.

*Figure 15.* Backend and frontend back-translation prompts for implementing a new project from an old codebase, used by the Trajectory Back-Translation Agent.

**Augmentation Prompts**

**Augmentation Planning Prompt**

You are a senior full-stack engineer who deeply understands website development (project structure, routings, server actions, and data layer).

Create five augmentation ideas for the current repository:
1. **Simplify** — The project remains a fully working full-stack site (API routes, database, auth, etc.) but becomes leaner and easier to maintain.
2. **Extend** — Introduce fresh user-facing functionality that brings additional value. Focus on features rather than swapping tech stacks.
3. **ParallelApp** — Invent three new products with different goals yet re-use the same code organisation so that major folders/components stay intact.

For inspiration (examples only — craft your own ideas):
• **Simplification:** remove dead code, unify state handling, strip unused tables, streamline data fetching, tighten typing.
• **Extension:** add booking workflow, user dashboards, advanced search, reporting.
• **Parallel Apps:** event ticketing portal, recipe catalogue, internal HR portal.

You MUST output a single JSON object having exactly the following top-level shape:
• `{ "augmentationPlans": [ /* array with 5 items, in the order below */ ] }`

Every element inside `augmentationPlans` must be an object with:
• `"name"`: string — short title of the proposal
• `"goal"`: string — what we want to achieve
• `"type"`: enum["simplify", "extend", "parallelApp"]
• `"keyChanges"`: array¡string¿ — 3–7 bullet points explaining main code changes / deletions / additions
• `"estimatedEffort"`: string — rough sizing like "XS / S / M / L / XL"
• `"expectedBenefits"`: string — why this plan is worth doing

The first element MUST have `"type": "simplify"`, the second MUST have `"type": "extend"`, and the third to fifth MUST have `"type": "parallelApp"`.

Nothing except the JSON is allowed in the final answer (no markdown fences).

**Example (illustrative):** [Example]

- - - - - - - - - - - - - - - - - - - - - - - - - - - - - - - - - - - - - - - - - - - - - - - - - - -

**Augmentation Implementation Prompt**

You are a principal software engineer with write access to the repository. Implement the **Augmentation Plan** below by directly editing the codebase with the provided file-system tools.

**Augmentation Plan:**
• **Augmentation Name:** [Name]

• **Augmentation Goal:** [Goal]

• **Augmentation Type:** [Type]

**Key Changes to be Made:**
• [Key Changes]
• ...

**Expected Benefits:** [Expected Benefits]

**General rules:**
1. All modifications must correspond one-to-one with the Key Changes bullets. Address them in the given order; if a bullet is ambiguous, clarify it in a short comment inside the code.
2. Use only the following tools to interact with the repo: `read_file`, `write_file`, `replace`, `list_directory`, `glob`, `search_file_content`, `run_shell_command`, `backend_test`, `frontend_test`.
3. Follow project conventions (code style, naming, lint rules, TypeScript strictness, existing folder layout, etc.).
4. Do not introduce completely new external services or dependencies not already present in `package.json` / `requirements.txt` unless the plan explicitly requires it.
5. After finishing modifications, run the relevant test tool(s) and fix any breaking issues you encounter.

Start by listing a high-level TODO checklist in the assistant message. Then iteratively execute those steps with the available tools until the augmentation is complete.

*Figure 16.* Prompts for the Augmentation Planning Agent and the Augmentation Implementation Agent.

Inspect the previous messages and verify whether **ALL** work described in the augmentation plan has been fully implemented in the present repository state.

**Augmentation Plan Under Test:**

- **Augmentation Name:** [Name]

- **Augmentation Goal:** [Goal]

- **Augmentation Type:** [Type]

**Key Changes to be Made:**

- [Key Changes]

- ...

**Expected Benefits:**

[Expected Benefits]

**Verification Procedure:**

1. Iterate over each bullet in the keyChanges array and determine the observable effect that change should have on the codebase.

2. Use **only** the following tools: {ReadFileTool.Name}, {ListDirectoryTool.Name}, {GlobTool.Name}, {GrepTool.Name}.

3. Maintain a checklist marking every bullet as *satisfied* or *unsatisfied*. A bullet is satisfied only if concrete code evidence is found.

4. If **ANY** bullet is unsatisfied, the entire augmentation is considered unsuccessful.

**Output Requirements:**

- Return a single JSON object with **exactly one field**.

- If all bullets are satisfied, output:

    – {"is_success": true}

- Otherwise, output:

    – {"is_success": false}

- Do **NOT** output anything else (no analysis, no markdown fences).

*Figure 17.* Prompt for verifying whether an augmentation plan has been fully implemented based on concrete code evidence.

**Frontend Testing Prompts**

**Starting Prompt:**

Now given a task:

**Task:**

Compare specific financial metrics between the second and the third companies using the dashboard's comparison feature.

**Expected Result:**

The comparison view displays both companies' financial metrics side-by-side, allowing for clear analysis and matching data sources.

**Instructions:**

- Attempt the task as a user would, using the UI elements available.

- Make multiple attempts if needed to try and achieve the expected result.

- Observe whether the expected result is fully, partially, or not at all achieved.

- **IMPORTANT:** You can at most interact with the website 15 times. If the limit is reached, directly output your answer.

At the end of your testing, answer only with one of the following:

- **YES**: if the expected result was fully achieved.

- **NO**: if the expected result could not be achieved at all.

- **PARTIAL**: if only some aspects of the expected result were achieved.

- - - - - - - - - - - - - - - - - - - - - - - - - - - - - - - - - - - - - - - - - - - - - - - - - - - - - - - - -

**Database Interaction Validation Prompt:**

You have just finished performing a GUI testing task based on the following task description and expected result:

**Task:**
`[GUI Testing Task]`

The GUI testing task may or may not require interactions with the database through the backend. The following are the database log entries that were written during the UI testing process:

**Database Logs:**
`[Database Logs]`

You should judge whether the database interactions that occurred during the UI testing process were appropriate and correct based on the above logs. Consider:

- Are the database operations consistent with what would be expected for the given UI testing task?

- Do the database operations support the functionality being tested?

- Are there appropriate database operations for actions such as form submissions, data updates, or searches?

- If the UI task involves viewing data, are there appropriate `SELECT` operations?

- If the UI task involves modifying data, are there appropriate `INSERT`, `UPDATE`, or `DELETE` operations?

- If the GUI task is purely navigation or viewing, it may not require database operations.

Answer based on the following criterion:

- If the GUI task does not require database operations, answer **YES**.

- If the database logs contain the necessary operations to support the testing task, answer **YES**.

- If the database logs do not contain the necessary operations to support the testing task, answer **NO**.

**Note:** Output the database interaction correctness judgement *exactly* in the following format:
`Database Interaction Correctness: [YES|NO]`

*Figure 18.* Frontend testing prompts, including GUI-agent testing and database interaction validation.

**Information Gathering Prompt**

You are a senior software developer good at gathering information about a codebase. You are tasked with gathering information about the backend APIs and database configuration from the codebase.

You need to gather the following information:

1. **Backend Port:** Find the port that the backend is listening on. Focus on configuration files and `.env` files in the backend directory.

2. **API Endpoints:** For each backend API endpoint, gather:

    - **Name:** A concise, descriptive endpoint name.
    - **Method:** HTTP method (e.g., GET, POST, PUT, DELETE).
    - **Path:** Full URL path of the endpoint.
    - **Description:** Brief explanation of the endpoint's behavior.
    - **Request Schema:** Structure of the request payload (fields and types).
    - **Response Schema:** Structure of the response payload (fields and types).
    - **Status Codes:** Possible HTTP status codes and meanings.

3. **Database Configuration:** Gather details including:

    - **Type:** Database type (e.g., PostgreSQL, MySQL, MongoDB).
    - **Database Path:** Path to database file or directory, if applicable.
    - **Connection Details:** Host, port, username, password, and database name using keys `db_host`, `db_port`, `db_username`, `db_password`, `db_name`.

Extract this information by examining the codebase, configuration files, and documentation. Use `read_file`, `search_file_content`, `glob`, and `list_directory` as needed.

**Important Notes:**

- If uncertainties arise, investigate further using the available tools.

- Ensure all gathered information is accurate and complete.

- If no API endpoints are found, set `api_endpoints` to an empty list. Do not fabricate endpoints.

**IMPORTANT:** The backend may define a global prefix (e.g., `/api`) in a global file such as `main.ts`. If such a prefix exists, it must be included in all API paths. For example, `/api` + `/posts` becomes `/api/posts`.

Your final output must be a single JSON object like the following example:
`[Example]`

Now begin gathering the information by exploring the codebase and using the tools as needed.

*Figure 19.* Prompt for gathering backend API and database configuration information from a codebase in preparation for backend tests.

**Backend Testing Prompts**

**Test Case Starting Prompt:**

You are a senior backend developer tasked with testing a backend API based on the user's instruction.

You will be given a backend testing task and the expected result. Your goal is to design and execute a backend API test that accurately reflects the user's intent and verifies the API functionality. You must use the specialized tool `backend_test`.

Follow these steps:

1. Analyze the user's instruction to identify the API endpoint, request method, parameters, and headers.

2. Formulate the exact testing input based on the codebase and instructions.

3. Execute the test using `backend_test`.

4. If the test fails due to possible misunderstanding of the API, investigate using `read_file`, `search_file_content`, `glob`, and `list_directory`, then adjust and re-run the test.

5. If the failure is due to invalid data, adjust the data and retry.

6. Some tests may require calling a registration API before testing login.

7. Output a final judgement:
   - **YES**: expected result fully achieved.
   - **NO**: expected result not achieved.

**Note:**
- If the failure is due to a genuine bug, do not re-run the test.

- Do not test the same API multiple times without new evidence.

- Output exactly: `Final Judgement: [YES|NO]`.

**Task:** [Task]
**Expected Result:** [Expected Result]

- - - - - - - - - - - - - - - - - - - - - - - - - - - - - - - - - - - - - - - - - - - - - - - - - - - - - -

**Database Interaction Validation Prompt:**

You have completed a backend testing process based on the following task and expected result:
**Task:** [Task]
**Expected Result:** [Expected Result]
**Database Logs:**
[Database Logs]
Judge whether the backend performed correct database interactions:
- If no database interaction is required, answer **YES**.

- If interaction is required, verify the presence of necessary operations:

  - `SELECT` for data retrieval.
  - `INSERT` or `UPDATE` for data creation or modification.
  - `DELETE` for data removal.
  - Ignore unrelated database operations.

- Output **YES** if required operations exist; otherwise output **NO**.

**Note:** Output exactly:
`Database Interaction Correctness: [YES|NO]`

*Figure 20.* Prompts for backend API testing and database interaction validation.

You are a meticulous software engineer specialised in databases.

Below is the implemented database schema (tables → columns with a few rows) dumped at runtime.

**IMPLEMENTED DATABASE SCHEMA:**
```json
[Database Schema in JSON]
```

Answer the following question:
Does the database contain `[Test Case Data Description]`?

Consider whether the implemented database contains a table (or a combination of tables) that can provide the content in the question. Answer `"Yes"` if:

- a single table matches, OR

- several tables together cover the purpose, OR

- a superset of a table matches.

Otherwise answer `"No"`.

**Steps for you:**

1. Think step-by-step. Show your reasoning for each question.

2. After finishing all reasoning, output ONLY one valid JSON object fenced in ```json ... ``` with the key `"answer"`.

   - If the answer is yes, output: {`"answer"`: `"Yes"`}
   - Otherwise, output: {`"answer"`: `"No"`}

*Figure 21.* Prompt for verifying whether the implemented database schema contains data required by a test case.

**Instruction:**
You are tasked with evaluating the functional design of a website that had been constructed based on the following instruction:
`{instruction}`

Grade the website's appearance on a scale of 1 to 5 (5 being highest), considering the following criteria:

- **Successful Rendering:** Does the website render correctly without visual errors? Are colors, fonts, and components displayed as specified?

- **Content Relevance:** Does the design align with the website's purpose and user requirements? Are elements (e.g., search bars, report formats) logically placed and functional?

- **Layout Harmony:** Is the arrangement of components (text, images, buttons) balanced, intuitive, and clutter-free?

- **Modernness & Beauty:** Does the design follow contemporary trends (e.g., minimalism, responsive layouts)? Are colors, typography, and visual hierarchy aesthetically pleasing?

**Grading Scale:**

- **1 (Poor):** Major rendering issues (e.g., broken layouts, incorrect colors). Content is irrelevant or missing. Layout is chaotic. Design is outdated or visually unappealing.

- **2 (Below Average):** Partial rendering with noticeable errors. Content is partially relevant but poorly organized. Layout lacks consistency. Design is basic or uninspired.

- **3 (Average):** Mostly rendered correctly with minor flaws. Content is relevant but lacks polish. Layout is functional but unremarkable. Design is clean but lacks modern flair.

- **4 (Good):** Rendered well with no major errors. Content is relevant and logically organized. Layout is harmonious and user-friendly. Design is modern and visually appealing.

- **5 (Excellent):** Flawless rendering. Content is highly relevant, intuitive, and tailored to user needs. Layout is polished, responsive, and innovative. Design is cutting-edge, beautiful, and memorable.

**Task:**
Review the provided screenshot(s) of the website. Provide a detailed analysis and then assign a grade (1–5) based on your analysis. Highlight strengths, weaknesses, and how well the design adheres to the specifications.

**Your Response Format:**
**Analysis:** [2--4 paragraphs addressing all criteria, referencing the instruction]
**Grade:** [1--5]
**Your Response:**

*Figure 22.* Prompt for evaluating website appearance from screenshots and assigning a 1–5 grade.

