# OpenReview forum: "FullStack-Agent: Enhancing Agentic Full-Stack Web Coding via Development-Oriented Testing and Repository Back-Translation"
_ICML.cc/2026/Conference — ICML 2026 regular_

### Official Review · Reviewer_fpVz · 2026-03-01

**Soundness:** 3
**Presentation:** 3
**Significance:** 3
**Originality:** 3
**Overall Recommendation:** 4
**Confidence:** 4

**Summary:**

This paper addresses an important practical gap in current coding agents: many systems appear able to “generate websites,” but in reality, they often only produce the frontend, without a functional backend or database support. To address this, the paper proposes a full-stack agent system consisting of: (1) a multi-agent development framework for full-stack website generation; (2) a self-improvement method based on GitHub repository back-translation and augmentation; and (3) a benchmark that jointly evaluates frontend, backend, and database functionality.

**Compliance With Llm Reviewing Policy:**

Affirmed.

**Final Justification:**

My concerns have been adequately addressed.

**Key Questions For Authors:**

1. Can this work be applied beyond website development, and if so, how? For example, could the framework be extended to other software development domains such as mobile applications or desktop applications?
2. Since the method augments data from existing repositories, it may also inherit built-in security flaws, vulnerabilities, or even backdoors. Does this require additional auditing or sanitization procedures?
Please also address the concerns raised in the **Weaknesses** above.

**Limitations:**

Yes.

**Strengths And Weaknesses:**

## Strengths
1. The paper focuses on a meaningful pain point: real website development requires not only UI generation, but also correct data flow, APIs, and database interactions.
2. The paper covers the full pipeline of method, training, and evaluation, and the problem formulation is clear.
3. The authors provide human analysis to support evaluation reliability, which is commendable.
4. The agent architecture is reasonably designed and well-motivated.

## Weaknesses
1. The maximum number of tool calls at inference is set to 400, and training uses 32 H800 GPUs. This suggests that the computational cost may be quite high (including time and token cost), yet the paper lacks sufficient comparison and analysis of whether these costs are acceptable in practice.
2. Some experimental results are not analyzed in enough depth. In particular, the proposed method outperforms the baselines on **FE Acc**, even though those baselines are primarily designed for frontend generation; this result deserves further investigation and explanation.
3. It is somewhat disappointing that the experiments only use Qwen as the backbone. Including other code-focused model families (e.g., Claude or Codex) would make the results more convincing.

---

> ### Author Rebuttal · Authors · 2026-03-29
>
> Thank you for your questions. We address your concerns below:
>
> **Q1.** The maximum number of tool calls at inference is set to 400, and training uses 32 H800 GPUs. This suggests that the computational cost may be quite high (including time and token cost), yet the paper lacks sufficient comparison and analysis of whether these costs are acceptable in practice.
>
> **A1.** The computational costs of FullStack-Dev with Qwen3-Coder-480B are presented in the table below. While the token count and runtime are slightly higher than those of baseline methods, they remain within an acceptable range. This is a necessary trade-off given the complexity of the full-stack development task, and the improved performance of our method justifies the additional computational and runtime costs.
>
> |Method|LLM Calls|Tools|Tokens|Runtime (min)|
> |-|-|-|-|-|
> |WebGen-Agent|12.0|-|71874.7|20.1|
> |TDDev|4.0|-|66460.4|17.9|
> |OpenHands|125.8|121.8|64802.2|27.8|
> |Bolt.diy|1.0|-|11187.5|7.6|
> |Qwen-Code|98.1|97.1|47145.9|16.5|
> |FullStack-Agent|197.4|190.7|85701.8|30.7|
>
> **Q2.** Some experimental results are not analyzed in enough depth. In particular, the proposed method outperforms the baselines on FE Acc, even though those baselines are primarily designed for frontend generation; this result deserves further investigation and explanation.
>
> **A2.** As explained in Sec. 3.1 and Appendix E, the baseline agents tend to generate only the frontend web pages when provided with user instructions alone, so we also explicitly prompt them to generate backend components. When generating both frontend and backend, the resulting websites tend to exhibit data-fetching or API communication errors, significantly impacting frontend functionality. The performance reported in their papers tends to be high because the webpages use mock data to mask the lack of real full-stack data processing and storage in the frontend implementation.
>
> **Q3.** It is somewhat disappointing that the experiments only use Qwen as the backbone. Including other code-focused model families (e.g., Claude or Codex) would make the results more convincing.
>
> **A3.** To show that our method can generalize to other models, we evaluated FullStack-Dev with DeepSeek-V3, an LLM chosen for its strong performance and cost-effectiveness. As shown in the table below, FullStack-Dev with DeepSeek-V3 significantly outperforms other agent frameworks across all metrics, strongly suggesting that the performance gains mainly come from our framework design rather than being tied to a specific backbone model.
>
> |Model-DeepSeek-V3|FE Acc.|FE Acc. w/ Valid DB|BE Acc.|BE Acc. w/ Valid DB|DB Acc.|Appearance|
> |-|-|-|-|-|-|-|
> |WebGen-Agent|48.8|44.4|64.2|59.6|71.7|3.33|
> |TDDev|41.3|39.3|61.3|57.0|72.5|2.28|
> |OpenHands|21.8|19.3|76.5|74.0|74.6|1.69|
> |Bolt.diy|25.4|23.0|62.6|32.1|65.3|2.11|
> |Qwen-Code|22.1|20.1|65.9|64.6|70.7|1.58|
> |FullStack-Dev|**73.6**|**69.6**|**91.1**|**89.7**|**79.2**|**4.02**|
>
> **Q4.** Can this work be applied beyond website development, and if so, how? For example, could the framework be extended to other software development domains such as mobile applications or desktop applications?
>
> **A4.** The FullStack-Agent system could potentially be extended to mobile and desktop application development by creating corresponding templates with the appropriate development tools and installing the necessary dependencies in the sandbox environment. However, because our paper primarily focuses on full-stack web development, these applications are beyond the scope of this work and represent an interesting direction for future research.
>
> **Q5.** Since the method augments data from existing repositories, it may also inherit built-in security flaws, vulnerabilities, or even backdoors. Does this require additional auditing or sanitization procedures?
>
> **A5.** Our paper introduces FullStack-Agent, a novel agentic system for full-stack development, with a primary focus on improving the success rate of implementing functionality, which is already a challenging task. As mentioned in the Impact Statement, errors or vulnerabilities in agent-generated websites pose potential risks, which could be mitigated through further targeted testing mechanisms. However, this is beyond the scope of this paper and would be an interesting direction for future research.

---

> > ### Author Rebuttal · Reviewer_fpVz · 2026-04-03
> >
> > I appreciate the response and effort in addressing the concerns. My concerns have been adequately addressed, so I will raise my score.

---

> > > ### Author Response · Authors · 2026-04-04
> > >
> > > Dear Reviewer fpVz,
> > >
> > > Thank you for your response and for recognizing our efforts in the rebuttal. We are grateful for your time and helpful comments, which have greatly improved our project.
> > >
> > > Sincerely,
> > > Authors

---

### Official Review · Reviewer_tLSA · 2026-03-10

**Soundness:** 3
**Presentation:** 2
**Significance:** 3
**Originality:** 3
**Overall Recommendation:** 4
**Confidence:** 3

**Summary:**

This paper introduces FullStack-Agent, a unified system with three components: FullStack-Dev for generation, FullStack-Learn for model improvement, and FullStack-Bench for evaluation. The work targets a key limitation of existing LLM web-coding agents, which often produce convincing frontends but lack correct backend behavior. Empirical results for this pipeline demonstrate improved performance on full-stack web generation tasks.

**Compliance With Llm Reviewing Policy:**

Affirmed.

**Final Justification:**

The rebuttal helps clarify the intended scope of the work and provides some evidence of flexibility across frameworks. Although evaluation as being conducted currently is still relatively simplified full-stack settings, the overall contribution still warrants a positive assessment.

**Key Questions For Authors:**

1. How sensitive is the learning pipeline and the resulting model performance to the specific template structures used in the framework? To what extent is the model learning general full-stack development patterns versus performing template completion within a fixed scaffold?
2. For the learning process, how different are the reconstructed trajectories generated through back-translation compared with trajectories generated directly by FullStack-Dev (given only the user instruction summarized in the initial information-gathering stage)? If they are similar, why is back-translation necessary? More generally, if I understand correctly, the paper suggests this process aims to mimic real development workflows (similar to repo commit histories in software engineering). Do the generated trajectories actually resemble such workflows? For example one would imagine iterative debugging and refactoring being necessary steps for complex development tasks.

3. It would be helpful if the authors could provide more details on the computational cost of the pipeline, e.g., number of tool calls, average generation time etc, as well as the practical reproducibility of the system.

**Limitations:**

Little discussion of the limitations, e.g., the focus on template-based CRUD applications and the computational cost.

**Strengths And Weaknesses:**

Strength:
1. The idea of full-stack generation is well motivated and important, and the results presented in the paper suggest promising performance improvement. This seems like a genuinely useful engineering contribution.
2. The benchmark is a valuable contribution because it addresses an evaluation gap in prior work that of allows models to fake interactivity without implementing actual backend behavior.

Weakness:
1. The writing of this paper could benefit from polishing. The fullstack-learn section is difficult to follow. In particular, it is unclear what the input/output trajectories actually look like and how complex the underlying repositories and reconstructed trajectories are. There are also a number of grammatical errors and awkwardly structured sentences that collectively make the reading experience less smooth.
2. The pipeline appears to focus primarily on simplified CRUD-style applications within fixed templates. It is unclear how well the approach generalizes to out-of-distribution stacks (e.g., authentication-heavy applications or systems with more complex architectures), so it is difficult to interpret the results as evidence of broader web engineering capability.

---

> ### Author Rebuttal · Authors · 2026-03-29
>
> Thank you for your questions.
>
> **Q1.** It is unclear what the trajectories actually look like and how complex they are. There are also grammatical errors and awkward sentences.
>
> **A1.** The trajectories consist of reasoning and tool calls, including tool calls for file navigation, file writing, and editing, as well as debugging and validation. The raw trajectories from back-translation also consist of these processes, except that the file contents are filled based on the original repo. These interactions with the original repo are removed from the raw trajectories, as described in Appendix C. The repos are crawled from GitHub and contain complex page layouts and mature functionality.
>
> We have double-checked the section and will make adjustments such as the following:
>
> L188:  "guide it through the process of transcribing" to "to guide the agent in transcribing..."
>
> L201: "the frontend and backend plans...as well as a user instruction" to "frontend and backend plans...and user instructions"
>
>
> **Q2.** The pipeline appears to focus primarily on simplified CRUD-style applications within fixed templates.
>
> **A2.** In our paper, we introduce FullStack-Agent, the first system to provide a unified agentic solution for full-stack website generation, evaluation, and model self-improvement. It demonstrates significant advancements over prior works [1, 2], which test only the frontend and overlook the lack of real backend logic. Our approach can be generalized to different web frameworks, as shown in Appendix B, demonstrating its flexibility and generalizability. It already covers a large portion of full-stack development scenarios; while there may be additional complexities in web development, they are beyond the scope of this work and represent an interesting direction for future research.
>
> [1] Lu, Zimu, et al. "Webgen-bench: Evaluating llms on generating interactive and functional websites from scratch."
>
> [2] Sun, Haoyu, et al. "FullFront: Benchmarking MLLMs Across the Full Front-End Engineering Workflow."
>
> **Q3.** How sensitive is the learning pipeline to the specific templates? To what extent is the model learning general patterns versus performing template completion?
>
> **A3.** To demonstrate that the model is not just learning template completion within a fixed scaffold, we evaluate FullStack-Learn-LM, which was trained on Next.js and NestJS, using Django and Vue. As shown in the table below, evaluation on frameworks not seen during training still yields notable improvements, suggesting that the model is learning general full-stack development skills such as debugging and refactoring.
>
> |Django+Vue|FE Acc.|FE Acc. w/ Valid DB|BE Acc.|BE Acc. w/ Valid DB|DB Acc.|Appearance|
> |-|-|-|-|-|-|-|
> |Qwen3-Coder-30B|44.1|37.7|52.8|46.4|60.2|3.01|
> |FullStack-Learn-LM|45.8|39.3|54.5|50.7|68.1|3.22|
>
> **Q4.** How different are the back-translated trajectories compared with directly generated ones? Why is back-translation necessary? Do the generated trajectories actually resemble real development workflows? Would it lack debugging and refactoring?
>
> **A4.** The reconstructed trajectories include file navigation, file editing, and debugging processes, similar to directly generated trajectories. However, they can also draw on expert implementations from crawled repositories, whereas directly generated trajectories rely entirely on the model's own knowledge. As shown below, we directly generated 2K trajectories from instructions summarized in the info-gathering stage. The model trained on this data performs significantly worse than the model trained on back-translated trajectories, demonstrating the importance of back-translation.
>
> |Data|FE Acc.|FE Acc. w/ Valid DB|BE Acc.|BE Acc. w/ Valid DB|DB Acc.|Appearance Score|
> |-|-|-|-|-|-|-|
> |Directly-Generated 2K|38.6|35.6|46.5|44.0|49.4|2.82|
> |Back-Translated 2K|49.3|42.3|55.6|45.4|51.2|3.32|
>
> We manually checked the back-translated trajectories and observed that most of them contain four to five frontend and backend debugging tool calls, as well as patterns of refactoring. This is because the complexity of the repositories often makes adapting them to the given template nontrivial. This resembles a human learner trying to reproduce an existing web project. The agent still needs to debug and refactor the codebase, thereby incorporating necessary development workflows.
>
> **Q5.** Provide more details on the computational cost.
>
> **A5.** The computational costs are shown below. While the cost is slightly higher than baseline methods, they are still within an acceptable range. This is a necessary tradeoff considering the complexity of the full-stack development task, and the improved performance justifies the extra cost.
>
> |Method|LLM Calls|Tools|Tokens|Runtime (min)|
> |-|-|-|-|-|
> |WebGen-Agent|12.0|-|71874.7|20.1|
> |TDDev|4.0|-|66460.4|17.9|
> |OpenHands|125.8|121.8|64802.2|27.8|
> |Bolt.diy|1.0|-|11187.5|7.6|
> |Qwen-Code|98.1|97.1|47145.9|16.5|
> |FullStack-Agent|197.4|190.7|85701.8|30.7|

---

> > ### Author Rebuttal · Reviewer_tLSA · 2026-04-03
> >
> > Thank you for the clarification. The rebuttal helps clarify the intended scope of the work and provides some evidence of flexibility across frameworks. Evaluation as being conducted is still relatively simplified full-stack settings, and broader generalization would benefit from further evidence. That said, this does not change my overall positive assessment.

---

> > > ### Author Response · Authors · 2026-04-04
> > >
> > > Dear Reviewer tLSA,
> > >
> > > Thank you for acknowledging our rebuttal and for your constructive feedback. We appreciate the time you invested and the insights you shared, which have helped us improve our work.
> > >
> > > Our paper introduces FullStack-Agent, a unified system for agentic generation and evaluation of full-stack websites, as well as for improving backbone LLMs. The system generalizes across different web-development frameworks and covers a wide range of web-development scenarios. While there may still be complications in web development that are not addressed, they are beyond the scope of this work and represent a promising direction for future research.
> > >
> > > Thank you again for your insightful comments and valuable suggestions.
> > >
> > > Sincerely,
> > > The Authors

---

### Official Review · Reviewer_ZQ9Z · 2026-03-12

**Soundness:** 3
**Presentation:** 3
**Significance:** 3
**Originality:** 3
**Overall Recommendation:** 4
**Confidence:** 4

**Summary:**

This paper addresses the challenge of enabling LLM-powered code agents to generate production-level full-stack web applications beyond simple frontend pages. The authors propose FullStack-Agent, which consists of three main components: (1) FullStack-Dev, a multi-agent framework with planning, code editing, codebase navigation, and bug localization capabilities; (2) FullStack-Learn, a data-scaling and self-improvement method that back-translates crawled and synthesized website repositories to enhance the backbone LLM; and (3) FullStack-Bench, a comprehensive benchmark testing frontend, backend, and database functionalities. The paper evaluates the approach on their benchmark, claiming substantial improvements over prior state-of-the-art methods: 8.7% on frontend tests, 38.2% on backend tests, and 15.9% on database tests. Additionally, they demonstrate that FullStack-Learn improves a 30B model's performance by 9.7%, 9.5%, and 2.8% on the three test categories through self-improvement.

**Compliance With Llm Reviewing Policy:**

Affirmed.

**Final Justification:**

The paper is technically solid and makes a useful contribution to the study of reasoning over real-world files. The authors’ rebuttal clarified several important details and improved the overall clarity of the submission.

**Key Questions For Authors:**

1. Has FullStack-Dev been evaluated on any existing benchmarks (e.g., WebArena, SWE-bench)? If not, what would be the expected results? Cross-benchmark performance would significantly strengthen generalization claims.

2. What proportion of back-translated pairs are filtered out during self-improvement? Can you provide examples of high-quality vs. low-quality generated instructions? How do you ensure filtering doesn't bias the task distribution toward simpler instances?

3. Can you provide pseudocode or detailed flowcharts illustrating how Planner, Editor, Navigator, and Commander coordinate? What are the specific trigger conditions and conflict resolution mechanisms?

4. What are the average LLM call counts, token consumption, and wall-clock time for FullStack-Dev compared to baselines? How do costs scale with task complexity?

**Limitations:**

yes

**Strengths And Weaknesses:**

### Strengths

1. The paper provides a clear problem formulation for complete full-stack development, including data flow control, dependency management, and cross-stack bug localization, moving beyond the limitation of merely generating frontend pages.

2. The back-translation pipeline that converts crawled repositories into instruction-response pairs, combined with pass@1-based self-improvement filtering, offers an innovative solution to the data scarcity problem.

3. Table 1 demonstrates substantial improvements of 38.2% on backend tests and 15.9% on database tests, making meaningful progress on previously neglected critical bottlenecks.

### Weaknesses

1. The paper evaluates exclusively on the self-constructed FullStack-Bench without any external benchmark or real-world deployment validation. The method design is highly coupled with the benchmark design, raising concerns about overfitting risk and severely insufficient evidence of generalization.

2. The paper lacks details on quality assurance for generated instruction-response pairs, diversity analysis, and filtering rate statistics. The pass@1-based filtering may introduce selection bias, and the quality control mechanism for repository back-translation is unclear.

3. While the paper describes four agents (Planner, Editor, Navigator, Commander), it lacks precise specifications of coordination protocols, decision-making processes, and invocation timing. This lack of specification directly impacts the work's reproducibility.

4. The paper does not report computational costs including LLM call counts, token consumption, or latency. Multi-agent systems typically incur higher overhead, requiring explicit cost-performance trade-off analysis to assess real-world deployment feasibility.

5. The proposed development-oriented test generator lacks quality validation: How is test correctness ensured? What are the false positive/negative rates? How does coverage compare to human-written tests? These critical questions remain unanswered.

---

> ### Author Rebuttal · Authors · 2026-03-29
>
> Thank you for your questions.
>
> **Q1.** Has FullStack-Dev been evaluated on any existing benchmarks (e.g., WebArena, SWE-bench)? If not, what would be the expected results?
>
> **A1.** FullStack-Agent is a specialized system designed for full-stack development, whereas WebArena is used for testing UI agents and SWE-bench is used for evaluating GitHub issue resolution; both are very different from the purpose of our system. To validate the generalizability of our method, we choose Web-Bench, a benchmark for evaluating the ability of web development. Compared to the baseline agent in [1], FullStack-Agent improves pass@1 accuracy by 3.6%, demonstrating the generalizability of our approach.
>
> |Method|Pass@1|
> |-|-|
> |Qwen3-Coder-480B + Baseline|16.4%|
> |Qwen3-Coder-480B + FullStack-Agent|20.0%|
>
> [1] Xu, Kai, et al. "Web-bench: A llm code benchmark based on web standards and frameworks." arXiv:2505.07473.
>
> **Q2.** What proportion of back-translated pairs are filtered out? Can you provide examples of high-quality vs. low-quality instructions? How do you ensure filtering doesn't bias toward simpler instances?
>
> **A2.** To ensure the quality of generated data, we construct well-designed rules based on debugging tool outputs, as explained in Appendix D. Around 40% of back-translated pairs are filtered out. To ensure that filtering doesn’t bias toward simpler instances, we generate trajectories up to eight times, stopping only when a successful trajectory is acquired or the limit is reached, thereby ensuring high coverage. High-quality instructions have clear functionality, while low-quality instructions are vague. Examples are shown below.
>
> High-quality: Please implement a website for organizing bookmarks. The website should allow users to browse a collection of bookmarks organized by categories, with search functionality to find specific resources.
>
> Low-quality: Please implement a cross-platform desktop application template using Tauri with Next.js. The application should be capable of running on Windows, Linux, and macOS.
>
> **Q3.** Can you provide pseudocode illustrating how Planner, Editor, Navigator, and Commander coordinate? What are the specific trigger conditions and conflict resolution mechanisms?
>
> **A3.** As explained in Sec. 2.1, the FullStack-Dev framework consists of a Planning Agent, a Frontend Coding Agent, a Backend Coding Agent, and a Debugging GUI Agent. Their interactions are described in detail in Sec. 2.1, Fig. 1, and Appendix H. Pseudocode is shown below:
> ```
> U->Plan:make JSON(planF,planB)
> Plan->BE:planB
> BE:build; run BEdbg(API request)
> if BEdbg fail -> BE:fix -> BEdbg
> else BE->FE:apiSummary
> FE:build; run FEdbg(GUI agent)
> if FEdbg error -> BE:fix -> BEdbg
> ```
> Trigger conditions: Planning runs at start. Backend coding runs after planning. Frontend coding runs after API summary. Debug tools run when coding agents invoke them.
>
> Conflict resolution: single-writer lock; adapt frontend based on backend APIs.
>
> **Q4.** What are the average LLM call counts, token consumption, and wall-clock time for FullStack-Dev compared to baselines? How do costs scale with task complexity?
>
> **A4.** The computational costs of FullStack-Dev with Qwen3-Coder-480B are presented in the table below. While the token count and runtime are slightly higher than those of baseline methods, they remain within an acceptable range. This is a necessary trade-off given the complexity of the full-stack development task.
>
> Among the samples tested, the most expensive sample used 217664 tokens, while the least expensive used 35934. The most expensive sample involved complex functionality such as searching, filtering, and viewing records, whereas the least expensive involved only entering numbers and computing. This demonstrates that computational cost scales effectively with task complexity.
>
> |Method|LLM Calls|Tools|Tokens|Runtime (min)|
> |-|-|-|-|-|
> |WebGen-Agent|12.0|-|71874.7|20.1|
> |TDDev|4.0|-|66460.4|17.9|
> |OpenHands|125.8|121.8|64802.2|27.8|
> |Bolt.diy|1.0|-|11187.5|7.6|
> |Qwen-Code|98.1|97.1|47145.9|16.5|
> |FullStack-Agent|197.4|190.7|85701.8|30.7|
>
> **Q5.** How is test correctness ensured? What are the false positive/negative rates? How does coverage compare to human-written tests? These critical questions remain unanswered.
>
> **A5.** The test correctness is ensured through detailed console logs, returned data, and the judgment of a strong coding LLM. We randomly sampled 100 test cases generated during inference and manually verified their correctness. The false-positive and false-negative rates are shown in the table below. Both metrics are relatively low for frontend and backend tests, indicating that the testing mechanism provides reliable feedback. To analyze coverage, we randomly sampled 50 agent trajectories and found that only four contained minor omissions of functionality in backend tests and only five in frontend tests, demonstrating the high coverage of the agent-generated tests.
>
> ||FP (%)|FN (%)|
> |-|-|-|
> |Frontend|2|7|
> |Backend|3|3|

---

> > ### Author Rebuttal · Reviewer_ZQ9Z · 2026-04-03
> >
> > Thank you for the rebuttal. The additional experiments and clarifications sufficiently address my main concerns, especially on generalization, system details, and evaluation reliability. I therefore consider my concerns resolved.

---

> > > ### Author Response · Authors · 2026-04-04
> > >
> > > Dear Reviewer ZQ9Z,
> > >
> > > Thank you for acknowledging our rebuttal. We truly appreciate your time and the constructive feedback you provided, which has been invaluable in improving our work.
> > >
> > > Sincerely,
> > > Authors

---

### Official Review · Reviewer_a8uW · 2026-03-13

**Soundness:** 3
**Presentation:** 3
**Significance:** 3
**Originality:** 3
**Overall Recommendation:** 4
**Confidence:** 3

**Summary:**

This paper introduces FullStack-Agent, which aims to generate full-stack web applications from NL instructions. It targets the current gap that existing code agents tend to only generate front end pages. It consists of three tightly knit components: (1)  FullStack-Dev, which is a multi-agent framework for code planning, editing, navigation, and bug localization, (2) FullStack Learn for data scaling and self-improvement, and (3) FullStack-Bench, a benchmark that tests front and backend via agent judges. Extensive experiments are conducted and show that FullStack-Dev outperforms previous methods with a large margin on backend metrics. FullStack-Learn also consistently improves Qwen3-Coder-30B.

**Compliance With Llm Reviewing Policy:**

Affirmed.

**Final Justification:**

The authors have provided enough details to address my main concerns. I am more confident in my assessment of this paper as a solid work with meaningful contributions.

**Key Questions For Authors:**

1. How sensitive are the FullStack-Bench results to the choice of judge model? If another model other than Qwen is used as the judge, will it result in any difference in scores or rankings?
1. What would be the reason of the incremental improvement on the database tests for FullStack-Learn?
1. What is the computational cost of FullStack-Dev inference?

**Limitations:**

yes

**Strengths And Weaknesses:**

**Strengths**

- **The problem is well motivated and significant.** It is a meaningful stride to develop full-stack agents that not only develop visually appealing frontend pages.
- The three coupled compoents (agents, training, benchmark) demonstrate a comprehensive system design.
- **Strong empirical results.** FullStack-Dev with Qwen3-Coder-480B-A35B-Instruct outperforms WebGen-Agent by 38.2% and 15.9% on backend and database tests, which is a substantial improvement.
- **Practical debugging tools:** The frontend debugging tool that monitors terminal/console errors and the backend debugging tool (similar to Postman-style API testing) are great contributions. Ablation studies show the effectiveness of both debugging tools.

**Weaknesses**

- **Limited analysis for FullStack-Learn.** It is only tested on Qwen3-Coder-30B. The improvement on the database tests (2.8%) is incremental compared to the other two. This may not be statistically significant considering the size of the benchmark. Some analysis into why such a discrepancy exists would be interesting. Will the results generalize to other model families?
- **Scope:** All experiments use only Next.js + NestJS, which raises questions about its generalizability. It feels slightly overstated for the claim of production-level full-stack deployment given the template choice and that websites may involve other complexities such as third-party integration.
- **Baselines:** there is no comparison with closed-source models like GPT and Claude. Adding one in the comparison would tell us more about how much improvement comes from the framework v.s. the backbone LLM.

---

> ### Author Rebuttal · Authors · 2026-03-29
>
> Thank you for your questions. We address your concerns below:
>
> **Q1.** What would be the reason for the incremental improvement on the database tests for FullStack-Learn?
>
> **A1.** The reason that improvement on the database tests is less pronounced than in the other two areas might be due to the fact that the training data is constructed from GitHub repositories, which do not always contain database configurations. Therefore, guidance on database design and setup is weaker compared to frontend and backend development. However, there is still a notable improvement of 2.8%, demonstrating that FullStack-Learn has a positive effect on the model’s database skills as well.
>
> **Q2.** Will the results of FullStack-Learn generalize to other model families?
>
> **A2.** To validate that FullStack-Learn generalizes to other model families, we train Qwen3-30B-A3B-Instruct on 2K back-translated samples. As shown in the table below, FullStack-Learn significantly improves performance on the Qwen3-30B-A3B-Instruct model as well, demonstrating the generalizability of our training method.
>
> |Qwen3-30B-A3B-Inst.|Frontend Acc.|Frontend Acc. w/ Valid DB|Backend Acc.|Backend Acc. w/ Valid DB|DataBase Acc.|Appearance Score|
> |-|-|-|-|-|-|-|
> |orig|9.3|8.8|10.8|9.3|21.9|1.57|
> |trained with back-trans data 2K|27.7|24.7|27.6|24.7|22.9|2.42|
>
> **Q3.** Scope: All experiments use only Next.js + NestJS, which raises questions about its generalizability.
>
> **A3.** To demonstrate that our pipeline can generalize to other frameworks, we have added Vue.js and Django, two popular web development frameworks. As shown in the table below and in **Appendix B** of the paper, increasing the number of frameworks results in similar or even slightly better performance across all metrics, demonstrating the generalizability of our method.
>
> |FE|BE|FE Acc.|FE Acc. w/ Valid DB|BE Acc.|BE Acc. w/ Valid DB|DB Acc.|Appearance Score|
> |-|-|-|-|-|-|-|-|
> |Next.js|NestJS|67.8|64.7|**83.4**|77.8|77.9|**3.72**|
> |Next.js, Vue.js|NestJS, Django|**70.6**|**68.1**|80.5|**78.5**|**81.0**|3.58|
>
> **Q4.** Baselines: there is no comparison with closed-source models like GPT and Claude. Adding one in the comparison would tell us more about how much improvement comes from the framework v.s. the backbone LLM.
>
> **A4.** To demonstrate that our method generalizes to other models, we evaluate FullStack-Dev with DeepSeek-V3, an LLM selected for its strong performance and cost-effectiveness. As shown in the table below, FullStack-Dev with DeepSeek-V3 significantly outperforms other agent frameworks across all metrics, strongly suggesting that the performance gains mainly come from our framework design rather than being tied to a specific backbone model.
>
> |Model-DeepSeek-V3|FE Acc.|FE Acc. w/ Valid DB|BE Acc.|BE Acc. w/ Valid DB|DB Acc.|Appearance|
> |-|-|-|-|-|-|-|
> |WebGen-Agent|48.8|44.4|64.2|59.6|71.7|3.33|
> |TDDev|41.3|39.3|61.3|57.0|72.5|2.28|
> |OpenHands|21.8|19.3|76.5|74.0|74.6|1.69|
> |Bolt.diy|25.4|23.0|62.6|32.1|65.3|2.11|
> |Qwen-Code|22.1|20.1|65.9|64.6|70.7|1.58|
> |FullStack-Dev|**73.6**|**69.6**|**91.1**|**89.7**|**79.2**|**4.02**|
>
> **Q5.** How sensitive are the FullStack-Bench results to the choice of judge model? If another model other than Qwen is used as the judge, will it result in any difference in scores or rankings?
>
> **A5.** To evaluate the sensitivity of FullStack-Bench results to the choice of judge model, we replaced Qwen3-Coder-480B with DeepSeek-V3. As shown in the table below, replacing the judge model does not significantly affect the scores. DeepSeek-V3’s results are consistently lower by about 1-2 points, and the rankings of the tested agents remain unchanged.
>
> |Method|judge model|Frontend Acc. w/ Valid DB|Backend Acc. w/ Valid DB|DataBase Acc.|
> |-|-|-|-|-|
> |FullStack-Agent|Qwen3-Coder-480B|69.6|89.7|79.2|
> |FullStack-Agent|DeepSeek-V3|67.5|87.1|78.4|
> |WebGen-Agent|Qwen3-Coder-480B|44.4|59.6|71.7|
> |WebGen-Agent|DeepSeek-V3|42.0|58.8|71.0|
>
>
> **Q6.** What is the computational cost of FullStack-Dev inference?
>
> **A6.** The computational costs of FullStack-Dev with Qwen3-Coder-480B are presented in the table below. While the token count and runtime are slightly higher than those of baseline methods, they are still within an acceptable range. This is a necessary trade-off considering the complexity of the full-stack development task, and the improved performance of our method justifies the additional computational cost and runtime.
>
> |Method|LLM Calls|Tools|Tokens|Runtime (min)|
> |-|-|-|-|-|
> |WebGen-Agent|12.0|-|71874.7|20.1|
> |TDDev|4.0|-|66460.4|17.9|
> |OpenHands|125.8|121.8|64802.2|27.8|
> |Bolt.diy|1.0|-|11187.5|7.6|
> |Qwen-Code|98.1|97.1|47145.9|16.5|
> |FullStack-Agent|197.4|190.7|85701.8|30.7|

---

> > ### Author Rebuttal · Reviewer_a8uW · 2026-04-04
> >
> > I am maintaining my positive score, and the rebuttal largely resolve my concerns. I have two remaining questions:
> >
> > 1. For A2, the generalizability experiment uses Qwen3-30B-A3B-Instruct, which is still within the Qwen model family.
> >
> > 2. The explanation for the incremental improvement on database tests is intuitive but not empirically verified. Provision of some statistics would strengthen this claim.
> >
> > Overall, I am still leaning towards a (weak) accept as this is solid work with meaningful contributions.

---

> > > ### Author Response · Authors · 2026-04-05
> > >
> > > Dear Reviewer a8uW,
> > >
> > > Thank you for acknowledging our rebuttal. We truly appreciate your time and the constructive feedback you provided, which has been invaluable in improving our work.
> > >
> > > We address your follow up questions as follows:
> > >
> > > **Q1.** For A2, the generalizability experiment uses Qwen3-30B-A3B-Instruct, which is still within the Qwen model family.
> > >
> > > **A1.** Qwen3-30B-A3B-Instruct is a general-purpose LLM, unlike Qwen3-Coder-30B-A3B-Instruct, which was specifically trained for coding tasks. This suggests that our method can generalize to non-code-specialized LLMs. Because our compute is limited and training at a context length of 131,072 is very memory-intensive, we can train only models with about 30B parameters or fewer. In evaluating these smaller models, we found that only the Qwen models can generate long, agentic coding trajectories with multiple sequential tool calls. Other smaller models, such as gpt-oss-20b, become confused after a few iterations on FullStack-Dev and fail to generate websites effectively; their accuracy drops to a very low value (see the table below). This makes them unsuitable for the self-improvement process in FullStack-Learn, which is why we chose Qwen3-30B-A3B-Instruct for the generalizability experiment. However, this does not imply that our method cannot generalize beyond Qwen. As shown in A4 and in the table below, DeepSeek-V3 also performs strongly on FullStack-Dev, further demonstrating the generalizability of our system and suggesting that FullStack-Learn may also transfer to other LLM families.
> > >
> > > |Model Name|FE Acc.|FE Acc. w/ Valid DB|BE Acc.|BE Acc. w/ Valid DB|DB Acc.|Appearance Score|
> > > |---|---|---|---|---|---|---|
> > > |gpt-oss-20B|3.9|2.2|1.7|1.2|5.4|0.93|
> > > |DeepSeek-V3|73.6|69.6|91.1|89.7|79.2|4.02|
> > >
> > > **Q2.** The explanation for the incremental improvement on database tests is intuitive but not empirically verified. Provision of some statistics would strengthen this claim.
> > >
> > > **A2.** To provide statistical support for our explanation, we randomly sampled 100 repositories containing backend projects and examined their backend implementations. The results are shown in the table below. About 32% of the repositories do not have a clearly identified database implementation. PostgreSQL, the database we used in our experiments, accounts for the second-largest share at 29%. The remaining 39% is divided among other types of database implementations. This demonstrates that database configuration information is not always present in repositories and does not always align with the database used in our experiments, which likely explains the incremental improvement on database tests. However, FullStack-Learn still has a positive effect on the model’s database skills, with an improvement of 2.8%.
> > >
> > > |Category|Ratio (%)|
> > > |-|-|
> > > |No Clear Database|32|
> > > |PostgreSQL|29|
> > > |MySQL|14|
> > > |MongoDB|13|
> > > |SQLite|7|
> > > |TypeORM (driver unclear)|5|
> > >
> > > Thank you once again for your insightful suggestions and valuable comments.
> > >
> > > Sincerely,
> > > The Authors

---

### Decision · Program_Chairs · 2026-04-30

**Decision:**

Accept (regular)

**Comment:**

This paper takes on an important problem and puts together a fairly complete package, including the agent framework, the learning pipeline, and the benchmark. Reviewers generally agreed that the direction is worthwhile and that the backend results are encouraging. The main issue is that the evidence is still too tied to the authors’ own setup. Across the reviews, the concern was limited generalization: the method and benchmark are closely intertwined, and it is still unclear how well the conclusions carry beyond this environment.